# The role of cover crops for cropland soil carbon, nitrogen leaching, and agricultural yields - A global simulation study with LPJmL (V. 5.0-tillage-cc)

Vera Porwollik[1,2], Susanne Rolinski[1], Jens Heinke[1], Werner von Bloh[1], Sibyll Schaphoff[1], Christoph Müller[1]

[1]Potsdam Institute for Climate Impact Research, Member of the Leibniz Association, P.O. Box 60 12 03, 14412 Potsdam, Germany
[2]Department of Agricultural Economics, Humboldt-University of Berlin, Unter den Linden 6, 10099 Berlin, Germany

*Correspondence to:* Vera Porwollik (verapor@pik-potsdam.de)

**Abstract.** Land management practices can reduce the environmental impact of agricultural land use and production, improve productivity, and transform cropland into carbon sinks. In our study we assessed the biophysical and biogeochemical impacts and the potential contribution of cover crop practices to sustainable land use. We applied the process-based, global dynamic vegetation model LPJmL5.0-tillage-cc with a modified representation of cover crops, to simulate the growth of grasses on cropland in periods between two consecutive main crops' growing seasons for near-past climate and land use conditions. We quantified simulated responses of agroecosystem components to cover crop cultivation in comparison to bare soil fallowing practices. on global cropland for a period of 50 years.

For cover crops with tillage, we obtained annual global median soil carbon sequestration rates of 0.52 and 0.48 t C ha$^{-1}$ yr$^{-1}$ for the first and last decades of the entire simulation period, respectively. We found that cover crops with tillage reduced annual nitrogen leaching rates from cropland soils by median of 39 % and 54 % but also the productivity of the following main crop by average of 1.6 % and 2 % for the two analyzed decades. Largest reduction of productivity were found for rice, modestly lowered for maize and wheat, whereas soybean yield revealed an almost homogenous positive response to cover crop practices replacing bare soil fallow periods. Obtained simulation results of cover crop with tillage practices exhibit a good ability of the model version to reproduce observed effects reported in other studies. Further, the results suggest that no-tillage is a suitable complementary practice to cover crops, enhancing soil carbon sequestration and the reduction of nitrogen leaching while reducing potential trade-offs with the main crop productivity due to their impacts on soil nitrogen and water dynamics.

The spatial heterogeneity of simulated impacts of cover crops on the variables assessed here was related to the time period since the introduction of the management practice as well as to environmental and agronomic conditions of the cropland. This study supports findings of other studies, highlighting the substantial potential contribution of cover crop practices to the sustainable development of arable production.

## 1 Introduction

The agricultural sector is challenged to provide more food, feed, and fuel to meet an increasing demand due to global human population dynamics as well as changes in diet composition (Alexander et al., 2017; Bodirsky et al., 2015; Godfray et al., 2010). Simultaneously, it is expected to consume fewer resources either by direct savings or

by increasing general efficiency of applied inputs (Lal, 2004a; Springmann et al., 2018). Agricultural production accounts for ~10 % (mean of the years 2007 to 2016) of the annual global anthropogenic greenhouse gas emissions,

including carbon (C) dioxide, methane from ruminant animals, as well as nitrous oxide emissions from crop production (i.e. fertilizer) and livestock rearing activities (Rosenzweig et al., 2020). Additional to the estimated $1.6 \pm 0.7$ PgC yr$^{-1}$ emissions from land-use change for the decade 2010-2019 (Friedlingstein et al., 2020), about 1 PgC yr$^{-1}$ of emissions can be attributed to harvest, grazing, and tillage on global cropland in the period since the year 1850 (Pugh et al., 2015). Cropland covers about 12 % of the global ice-free land surface (Ramankutty et al.,

2008). A loss of 30 to 40 % soil organic C was estimated due to the historic cultivation of croplands (Poeplau and Don, 2015). At the same time, agricultural land management practices can be employed to reduce or reverse detrimental environmental impacts of agricultural production as well as facilitate the regeneration of degraded ecosystem services and functions (Rosegrant et al., 2014). Conservation Agriculture (CA) practices have been proposed to improve cropland soil fertility and to sustain productivity (Scopel et al., 2013; Thierfelder et al., 2018;

Tittonell et al., 2012). CA comprises minimum mechanical soil disturbance, the maintenance of a permanent vegetative soil surface cover, and a diversified crop rotation (Kassam et al., 2019). The latter two aspects can be accomplished by the integration of a secondary crop, which depending on the position and purpose in the rotation, can be referred to as green manure, intercrop, or as intermediate, companion, catch, and cover crop (term further used in this study). For farming systems cultivating annual crop types, cover crops can be grown between two

consecutive main cropping seasons, whereas for perennial woody crops, cover crops are rather found as groundcovers between trees (Gonzalez-Sanchez et al., 2019).

Cover crops exhibit several environmental benefits, such as decreasing nitrogen (N) leaching from agricultural systems (Abdalla et al., 2019; Thapa et al., 2018; Tonitto et al., 2006; Valkama et al., 2015). The N recovery rate of excess fertilizer left in the soil after harvest of a main crop is found to be higher for non-leguminous cover crop

species, such as grasses (e.g. ryegrass) and crucifers (e.g. radish) (Florentín et al., 2011) than for leguminous (e.g. peas and beans) cover crop species (Dabney et al., 2011; Valkama et al., 2015). Leguminous cover crop species are able to improve the N balance of the soil (Kaye and Quemada, 2017) through additional N fixation and by this way can reduce fertilizer input requirements in the long term (Nouri et al., 2020; Thierfelder et al., 2018). Last but not least, cover crops constitute a suitable measure for weed control and against soil compaction (SARE, 2019),

as well as erosion prevention through extending the vegetative coverage of the soil surface (Kaye and Quemada, 2017). Cover crops are terminated either naturally (e.g. by frost), chemically (e.g. by herbicide application), or mechanically (e.g. by mowing, roller, or tillage) (Kaye and Quemada, 2017). The corresponding biomass of the cover crops can be harvested for off-field usages, grazed by livestock, or if left on the field, be used to build up the soil's humus layer (Florentín et al., 2011). Cover crops are an important practice to manage soil fertility and

weed in organic farming systems (Keestra et al., 2018).

According to the Farm Structure Survey and the Survey on Agricultural Production Methods (SAPM), which are carried out on a 10 year interval as a census in the EU-28 countries, the soil surface of arable land during winter of the year 2010 was covered: 44 % with normal winter crops, 5 % with cover or intermediate crops, 9 % with plant residues, and 25 % left as bare soil. The remaining 16 % missing reporting share comprises areas under glass

(greenhouses) and areas not sown or cultivated during the reference year (e.g. temporary grassland) (EUROSTAT, 2018). Poeplau and Don (2015) report that current shares of cropland with cover crop cultivations range between 1-10 % in the US and for countries in Europe. Further, these authors estimate 25 % (~400 million hectares) of

cropland suitable for cover crop practices as half of the global winter or off-season fallow cropland, by excluding 50 % of the total area covered with winter cereals and further 25 % of the off-season fallow area due to climatic
or agronomic constraints. This area estimate is also used in Kaye and Quemada (2017), who find the mitigation potential of cover crop practices mainly due to the combined effects of soil C sequestration, reduced fertilizer application rates, and changes in surface albedo, corresponding to an off-set of about 10 % of the estimated annual emissions from agriculture. Cover crop practices encompass potential to contribute to climate change impact mitigation through soil C sequestration (Abdalla et al., 2019; Corsi et al., 2012; Poeplau and Don, 2015). Largest
potentials for the realization of C sequestration on global cropland soils were identified for areas with high natural potential soil C stocks and with strongest C depletion due to duration and intensity of historical agricultural land use and management (Sommer and Bossio, 2014), resulting in a larger saturation deficit (West and Six, 2007). Further, cover crop practices can serve adaptation and increasing the resilience of cropland production to climate change impacts through improving soil nutrient and water dynamics (Kaye and Quemada, 2017; Rosenzweig et
al., 2020). Dynamic global vegetation and land surface models can be used to assess impacts of land management practices on carbon, nitrogen, and water dynamics, across various temporal and spatial scales (Erb et al., 2016; McDermid et al., 2017; Pongratz et al., 2018). Hirsch et al. (2018) find considerable local temperature cooling effects in response to simulated Conservation Agriculture practices using the spatial explicit CA area dataset for the year 2012 by Prestele et al. (2018). However, assessments of the global carbon and other biogeochemical
cycles are hampered by the limited availability of data on cropland management practices at sufficient spatial and temporal resolution as well as level of detail captured by individual models (Pongratz et al., 2018). As a result, in global C cycle modelling assessments, 'cropland' often is represented as an aggregated effect across crop types and associated land management over large areas (Morais et al., 2019). Changes in management often can only be assessed via stylized model scenarios with homogenous assumptions on management intensities, or are restricted
to point scales simulations, for which more details on cropland management practices may be available (Lutz et al., 2020). Olin et al. (2015) explored the soil C sequestration potential of no-tillage, retaining main crop residues on the field, cover crops, and manure application for historical, current, and future climate simulation periods on cropland at the global scale using the process-based dynamic vegetation model LPJ-GUESSS. These authors found soil carbon sequestration with all alternative management scenarios compared to their standard simulation.
Additionally, for the cover crop scenario Olin et al. (2015) found reduced nitrogen (N) leaching rates by 15 % but also lowered main crop yields by 5 %, revealing a trade-off between agroecosystem services and functions.

Lutz et al. (2019) find a soil C sequestration potential within their simulated idealized no-tillage scenario but only when retaining all main crop residues on the field. However, findings by Herzfeld et al. (2021) reveal that with future climate change conditions, a switch to no-tillage, independent of the main crop residue removal rate, is not
sufficient to reverse projected soil carbon density declines on global cropland due to biomass extraction, conventional cropland management practices, and associated soil carbon decomposition processes.

The 'intercrop' carbon-only version of LPJmL and 15 other agroecosystem models were included in the study of Kollas et al. (2015) They find only minor ability of the model ensemble, comprising 15 models and including a preceding carbon-only LPJmL model version, to reproduce the slight positive main crop yield effect, which was
observed in the experimental site for the rotations with intermediate crops. It is important to understand the effects of cover crop practices on the terrestrial C and N cycles to improve model representation of the practices to be included in agricultural assessments. Therefore, it is the aim of our study to quantify the biophysical and

biogeochemical impacts and potential contribution of cover crop cultivation to sustainable arable production at the global scale accounting for differences in environmental and socio-economic conditions. We focus our analysis on effects of herbaceous cover crop species, growing as annual grasses and replacing bare soil fallows on cropland during main crop off-season periods. The objectives of this study were to: i) Assess the temporal and spatial pattern of cover crop cultivation impacts simulated with LPJml5.0-tillage-cc on global cropland soil C stocks, N leaching rates, and agricultural productivity, ii) Quantify responses to cover crop cultivation with regard to tillage practices and the influence of management duration, and iii) Estimate impacts of land management for the historical CA area and the potential contribution of cover crop practices to agricultural production impact and greenhouse gas mitigation efforts .

## 2 Methods and data

### 2.1 Model code functions in LPJmL5.0-tillage-cc

For the assessment of cover crop cultivation impacts we applied the dynamic global vegetation model LPJmL5.0-tillage-cc, representing biophysical and biogeochemical processes of the biosphere for the quantification of human-nature interactions as well as of their effects on natural and managed ecosystems. A detailed description of water, soil, and vegetation dynamics of a preceding carbon-only model version 4, including a comprehensive evaluation of model performance, is provided in Schaphoff et al. (2018a); (2018b). The here used model version additionally includes processes associated to global N dynamics in soils and plants (von Bloh et al., 2018) and an explicit representation of tillage and crop residue management (Lutz et al., 2019).

In the model three litter layers and five hydrologically active soil layers of differing thickness to a total depth of three meter are distinguished. Each soil layer has its specific temperature and moisture levels, affecting the decomposition rates of soil organic matter, represented in the model by fast and slow decomposing (30 and 1000 years turnover time, respectively) C and N pools (Lutz et al., 2019; Schaphoff et al., 2018a). Carbon and N pools of represented vegetation, litter, and soil layers are updated daily. Biomass formation is represented by a simplified version of photosynthesis according to Farquhar et al. (1980). The phenology of tree and grass plant functional types (PFTs) of the represented natural vegetation are based on Jolly et al. (2005) with modification of the growing season index as described in Forkel et al. (2014). Crop functional types (CFTs, see Table S1.1), representing the vegetation on cropland, are parameterized with specific temperature and phenological heat unit requirements for growth (Müller et al., 2017).

Cropland irrigation was mechanistically simulated by either surface flooding, sprinkler, or drip irrigation, here setting one type per country (Jägermeyr et al., 2015; Rohwer et al., 2007). We used the potential irrigation setting to simulate irrigated cropping systems (for cropland within grid cells with areas equipped for irrigation as informed by the input data (see Sect. 2.2) to account for missing representation of ground water sourcing, when this model version only considers surface water withdrawal amounts, in the case of alternatively setting to limited irrigation. The C to N ratio of manure was set 14.5 to 1. Half of the N contained in the manure was assumed as ammonium ($NH_4$) and added to the pool of the upper soil layer, whereas the entire C and the remaining N (assumed as organic share), were transferred to the respective litter pools. Generally, mineral N fertilizer and manure were applied to cropland at the sowing date of an individual main crop (CFT) within a grid cell. If the sum of N from the mineral N fertilizer and from the manure exceeded the threshold value of 50 kg N ha$^{-1}$, the remaining mineral N fertilizer

amount was applied at a second event during the growing season, when 40 % of the phenological heat sums to reach maturity, were accumulated.

Conventional tillage was assumed as the default historical soil management on all cropland, applied when converting land to cropland, as well as at main crop seeding and harvest events. After harvest of the main crop the tillage routine submerges and transfers 95 % of the aboveground biomass remaining on the field from soil surface to the incorporated soil litter pools. In the model, tillage mostly affects processes in the first soil layer up to 20 cm depth (Lutz et al., 2019). In the case of no-tillage, the remaining aboveground biomass of the main crops' residues left on the field after harvest are added to the surface soil litter pools, representing mulching practices. Herzfeld et al. (2021) examine global soil carbon dynamics affected by historical land use, land-use change, tillage, and crop residue management, based on simulations with a similar model code version, input data, and cropland management representation but different simulation setup than applied here. For the simulated period 2000-2009, the authors found a global cropland soil carbon stock of 170 PgC in response to historical dynamic climate input data, land use change, cropland use, and management practices, which was in good agreement with estimates reported in the literature.

## 2.2 Simulating cover crop practices with LPJmL5.0-tillage-cc

We used LPJmL5.0-tillage-cc with a modified code for the representation of cover crop management. It is built on an earlier version of the model accounting for 'intercrops', as the options to simulate either herbaceous vegetation, or bare soil fallow dynamics on cropland in periods between two consecutive main crops' growing seasons (Bondeau et al., 2007). The functionalities make use of three 'grass' plant functional types (PFTs) already implemented in LPJmL for the natural vegetation, growing on fallow cropland according to their bio-climatic limits as tropical C4, temperate C3, and polar C3 grass (Forkel et al., 2014). In the model, biophysical and biogeochemical dynamics on off-season cropland within a grid cell, are accounted for in routines of the 'setaside stand', preserving the separation of processes in soil columns into rainfed and irrigated shares.

As a first step, we modified the functionalities for the establishment of cover crop (grass), so that it occurs on each crop specific off-season cropland fraction after harvest of the main crop (CFT) within a grid cell. The initial biomass of the cover crop grass sapling (0.05-0.07 g C m$^{-2}$) was changed to be taken from the respective C and N pools of the soil litter layers. We did so, to avoid imposing artificial fertilization effects (Olin et al., 2015), from simply adding contained amounts of the sapling's C and N to the simulated system with the default CFTs establishment model routines, which assume crop seeds as external inputs.

In this model version, C and N are allocated to the different organs (root and leaf pools) of the cover crop grass plants on a daily basis, using routines of 'managed grassland' dynamics described in Rolinski et al. (2018) and von Bloh et al. (2018). Any management of the cover crops on fallow cropland was excluded, so that they were growing as grasses under rainfed conditions. Cover crops are terminated at the beginning of the following main crop growing season. The corresponding aboveground grass plant biomass is either left at the soil surface, or transferred to the incorporated soil litter pools, depending on the tillage setting. The root biomass of the terminated cover crops is added to the respective belowground litter pools. Soil and vegetation C, N, and water fluxes in the main crop growing period as well as during vegetated or bare fallow off-season were summarized in model outputs for the entire cropland. More details of the model functionalities and input data used are provided in the Supplement (Sect. S1).

## 2.3    Model input data

For the simulations of this study, the model was driven with monthly mean temperature input data from the Climate Research Unit (CRU TS version 3.23, University of East Anglia Climate Research Unit, 2015; Harris et al. (2014); covering the period 1901-2014). Monthly precipitation and number of wet days data were from the Global Precipitation Climatology Centre (GPCC Full Data Reanalysis version 7.0; Becker et al. (2013); years 1901-2013). The monthly radiation data (shortwave and net longwave downward) was taken from the ERA-Interim dataset (Dee et al., 2011) for the years 1901-2011. Annual atmospheric $CO_2$-concentration input data were based on the NOAA/ESRL Mauna Loa station reports (Tans and Keeling, 2015), and natural N deposition data on the ACCMIP database (Lamarque et al., 2013) for the years 1841-2012. Soil texture classes remained static over the simulation period and were based on the Harmonized World Soil Database (Nachtergaele et al., 2009) and soil-pH was taken from the WISE dataset (Batjes, 2006).

Model input data on historical land use, distinguishing shares of irrigated and rainfed crop-group specific physical cropland per grid cell, as well as mineral N fertilizer application rates were based on LUH2v2 data by Hurtt et al. (2020). The original data per crop group was (dis-)aggregated and remapped, using the MADRaT tool (Dietrich et al., 2020), to match the crop functional types (CFTs) representing vegetation dynamics on managed land in LPJmL (Table S1.1) and the here targeted model simulation grid cell resolution of 0.5 arc degree (~50 x 50 km at the equator). In the year 2010 there were ~1,500 million ha total global physical cropland (Fig. S1.2).

Sowing date and phenological heat units were prescribed with a growing season input dataset based on Portmann et al. (2010) and Sacks et al. (2010), described by Elliott et al. (2015). The historical manure input data was based on the time series of N contained in manure applied on cropland by Zhang et al. (2017). The residue management model input dataset prescribed the fraction of residue biomass remaining on the field after harvest of the main crop. It was generated, by setting residue recycling shares to values per CFT-group (i.e., cereals, fibrous, non-fibrous, and others), which were obtained from Dietrich et al. (2020) and based on national reported cropland data retrieved from FAOSTAT. The data accounts for historical main crop residue removal rates associated to land management practices, such as burning on field, as well as to secondary off-field usages, such as household burning, and livestock fodder.

## 2.4 Simulation setup of land management scenarios

As a first step, we conducted a 7000 years spin-up simulation with LPJmL5.0-tillage-cc, in order to get natural vegetation pattern and soil pools into a dynamic equilibrium state, recycling the first 30 years of climate input data following the procedures described in von Bloh et al. (2018). Subsequently, we ran a second spin-up simulation, with fixed cropland distribution pattern and most of the land management as provided by the model input data for the year 2010 (Sect. 2.3). We assumed bare soil fallow on cropland during the main crops off-season periods as well as tillage to be the default historical management practices. By keeping land use and management constant during this simulation step, we aimed to establish an equilibrium state between the C and N pools and the fluxes. We assumed that cropland had been already cultivated for a longer time at the beginning of the actual management simulation period so that our results can be more easily compared to literature values e.g., obtained from experiments conducted on already established cropland plots. Starting with cropland soil pools from this spin-up procedure, we simulated the control as reference scenario (REF) for 50 years of the historical period to present day, maintaining land use pattern and all land management model settings as during the land use spin-up period.

By using dynamic climate and $CO_2$ model forcing data during the actual management simulation period (years 1962-2011), we aimed to mimic near-past environmental production conditions. Three alternative cropland management scenario simulations were generated with: cover crops replacing bare soil fallow periods (CC), no-tillage (NT) applied as single, as well as combined cover crop and no-tillage practices (CCNT) on global cropland for the same 50 year simulation period and all other model settings as used in the reference scenario (REF) (see Supplement Table S1.3 for more details on simulation setup). On the one hand, this 50 year time frame has been chosen for analysis because it is stated as minimum duration required to re-establish a new steady state in soil C pools after the introduction of a new soil management practice involving altered biomass input levels to soils (Kaye and Quemada, 2017; Poeplau and Don, 2015). On the other hand, the 50 years were chosen for analysis because of spanning the maximum duration found for values reported in the literature and used here for evaluating simulated responses to cover crop practices (Table 1, Table S2.6).

## 2.5 Post-processing model outputs

Model output data was post-processed and analyzed with R version 3.3.2 (R Development Core Team, 2016), applying functions developed by Kowalewski (2016) as well as by using the packages 'raster' (Hijmans and van Etten, 2012), 'reldist' (Handcock, 2016), and 'ncdf4' (Pierce, 2015).

Soil C stock change was quantified up to a 30 cm soil depth by adding C pool model outputs for the litter, the first soil layer (0-20 cm soil depth), and one third of the second soil layer (20-50 cm soil depth). Responses of cropland soil C stock to altered management scenario (CC, CCNT, and NT) in comparison to the control (REF) were generated, assuming a 'paired plot' (West et al., 2004) or 'synchronic' approach (Corbeels et al., 2018). The calculations follow the equation 3.3.4B of the guidance from the Intergovernmental Panel on Climate Change (IPCC, 2006) for annual changes in mineral soil C stock on remaining cropland as Eq. (1):

$$\Delta p_{s,i,t} = (p_{s,i,t} - p_{REF,i,t})/T_{i,t} \, , \tag{1}$$

where $\Delta p_{s,i,t}$ is the annual soil C sequestration rate in t C ha$^{-1}$ yr$^{-1}$ per alternative scenarios $s$, in grid cell $i$, and time step $t$, as the absolute difference between the annual absolute soil C stock $p_{s,i,t}$ in t C ha$^{-1}$ yr$^{-1}$ in each of the alternative scenarios and the baseline $p_{REF,i,t}$, divided by management duration $T$, as the number of years (1 to 50) since introduction of the alternative practices.

Whereas all twelve CFTs were modeled (see S1.1), we focus our analysis on impacts of cover crop practices replacing bare soil fallow periods on the productivity of the following main crops of wheat, maize, rice, and soybean because of their global relevance as stable crops and their large spatial cropland coverage. Throughout the study we report main crop productivity impacts due to changes in management on each of the four main crop types' separated for irrigated and rainfed cropping systems or as changes of average productivity, as area-weighted mean of simulated irrigated and rainfed yields in kg DM ha$^{-1}$ yr$^{-1}$, per crop-specific cropland in grid cell $i$, and time step $t$. For area-weighting the model output data at the grid cell scale, we employed the crop-specific rainfed and irrigated cropland shares, which were used as land use model input data for the year 2010 (see Sect. 2.3).

Responses to simulated altered management of crop-specific yield in kg DM ha$^{-1}$ yr$^{-1}$ and cropland soil N leaching rates in kg N ha$^{-1}$ yr$^{-1}$, respectively, were computed as Eq. (2):

$$\Delta v_{s,i,t} = ((\frac{v_{s,i,t}}{v_{REF,i,t}}) - 1) * 100 \, , \tag{2}$$

where $\Delta v_{s,i,t}$ is the relative difference in percent (%) between the assessed variable ($v_{s,i,t}$) per alternative management scenario $s$ compared to the baseline value ($v_{REF,i,t}$), per hectare of cropland area in grid cell $i$, and time step $t$.

We report global aggregates of simulated values and differences as area-weighted median (Q2 as q = 0.5 as $\Delta \tilde{v}_{s,i,t}$), the first (Q1 as q = 0.25) and third quartiles (Q3 as q = 0.75) per management scenario $s,$ and per time step $t$. Time step $t$ is annual (yr$^{-1}$) either reported for the first (years 1 to 10), or last (years 41 to 50) decade of the 50 simulation years, in order to contrast short from long term management effects, or for an else indicated time period. For area-weighting of global aggregated changes in soil carbon and N leaching rate, we applied the sum of the physical cropland per grid cell using the land use data of the year 2010 (see Sect. 2.3, S1.2).

For evaluating LPJmL5.0-tillage-cc model results, we compare modeled responses to cover crop cultivation on soil carbon, nitrogen, and water dynamics to values reported in the literature, which use bare soil fallowing practices and conventional tillage in the control treatment (Table 1). Meta-analyses and reviews on cropland management effects summarize experimental studies' findings, covering a variety of temporal scales and crop production conditions regarding climate, soil, and management intensities. Although many studies present averages across experiment sites and years (Nyawira et al., 2016), we computed spatial and temporal aggregated median (and quartiles) changes to exclude outliers stronger influence on global spatial aggregated mean values. Further, we report crop productivity impacts of changes in cropland management as mean across aggregated yield change values obtained for each of the assessed four following main crop types, when a variety of main crop types were included in experiments of the literature value and used for comparison.

To assess historical global impact of Conservation Agriculture on soil C, N leaching rate, and main crop productivity, we employed a time series dataset of CA area of annual global gridded physical cropland covering the years 1974-2010. During the assessed historical period the global CA area grew from a share of 0.2 to 10 % of the global cropland (FAO, 2016). This CA dataset was generated by using annual national reported CA cropland data in hectares (FAO, 2016), and by employing downscaling methods described in Porwollik et al. (2019) as well as further in the Supplement (Sect. S1.4). The simulation cover crops combined with no-tillage (CCNT) was assumed a proxy for the full suite of CA practices, whereas responses to the no-tillage (NT) and cover crop with tillage scenario (CC) comprise only one single land management component of the principles promoted under CA, respectively. Computed changes per variable, grid cell $i$, time step $t$ for the CC, CCNT, and NT scenarios compared to the control (REF), were remapped to match the historically evolving spatial and temporal pattern of the CA area time series data. We quantified impacts of switching to single cover crop (CC), no-tillage (NT), and combined alternative cropland management practices (CCNT) on variables as global aggregated total and as area-weighted median change per hectare of CA cropland for the years 1974 to 2010.

## 3   Results

### 3.1 Soil carbon responses to altered land management and duration

We found increased cropland soil carbon stocks in the three simulated alternative land management scenarios compared to the control (REF), indicated by positive annual area-weighted spatial aggregated median soil carbon sequestration rates (Fig. 1, for respective spatial patterns see Fig. S2.1.1). During the first decade of the 50 year simulation period the median soil C sequestration rates in the three alternative management scenario simulations

CC, CCNT and NT were higher (0.52, 0.72, and 0.08 t C ha$^{-1}$ yr$^{-1}$) than during the last decade (0.48, 0.54, and 0.01 t C ha$^{-1}$ yr$^{-1}$) (Table 1, Table S2.2). The maximum annual median soil C sequestration rates within both cover crop scenarios CC and CCNT (0.79, 1.03 t C ha$^{-1}$ yr$^{-1}$) were reached in the sixth year of the analyzed 50 year simulation period, whereas in NT (0.11 t C ha$^{-1}$ yr$^{-1}$) already in the third year since introduction of altered management. After these peaks within each of the scenarios, the annual soil C accumulation effect persist over the course of the remaining simulation period, but with lower rates.

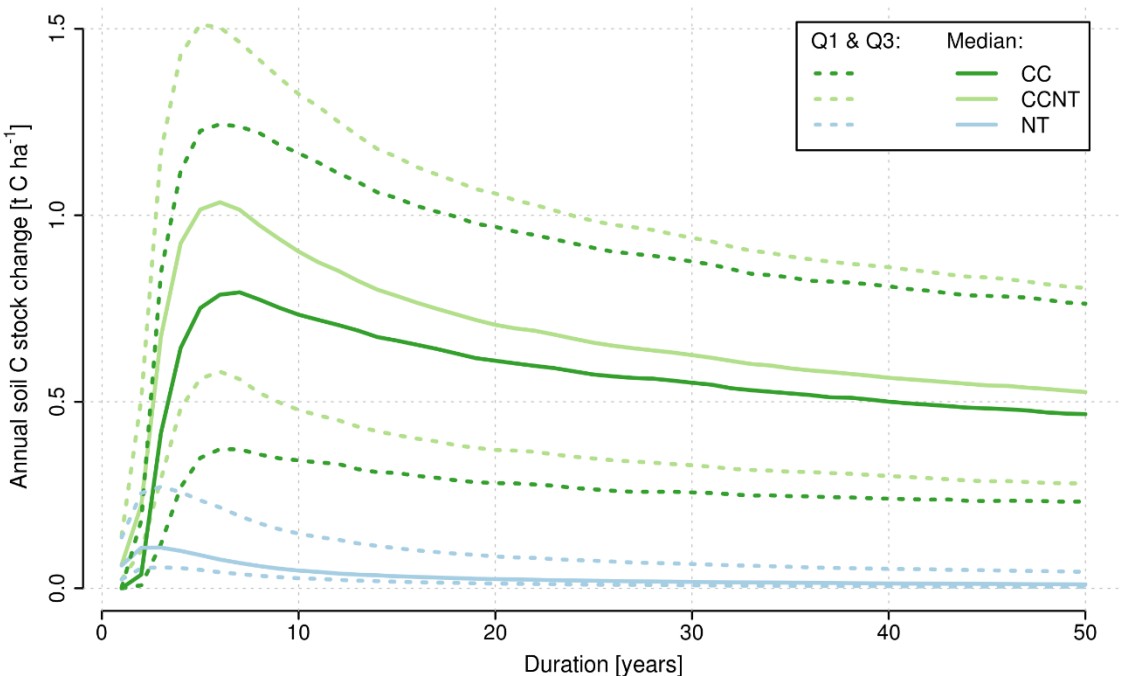

**Figure 1** Aggregated area-weighted median across global cropland (~1,500 million ha) of average annual soil C sequestration rates (Eq. 1) in t C ha$^{-1}$ yr$^{-1}$ as solid lines and the first (Q1) and third (Q3) quartiles as dashed lines per alternative land management scenario (CC: dark green, CCNT: light green, NT: light blue) compared to the baseline (REF) over the 50 year simulation period.

**3.2 Simulated impacts of land management on soil N and water dynamics**

All three alternative management scenarios exhibited higher transpiration but lower evaporation rates than found in the baseline (Fig. 2 a and b). In both cover crop simulations (CC and CCNT) the transpiration rates were higher because of the extended vegetative growth per cropland area unit compared to scenarios with the bare soil fallow during primary crop off-season periods (REF and NT). With CC, transpiration increased more strongly than evaporation was reduced, so that total evapotanspiration water fluxes were higher than in REF. In CCNT and NT, we found lowerded evaporation rates outweighing elevated transpiration rates compared to in REF with tillage. Cover crops in CC and CCNT led to lower but still positive median N net-mineralization rates (as the difference of soil N gross mineralization and immobilization rates) compared to bare soil fallowing practices in REF and NT (Fig. 2 c). This decline was driven by larger increases of the soil N immobilization than of gross mineralization rates, especially within the first 10 years after introduction of cover crop practices (Fig S2.3). In both cover crop scenarios (CC and CCNT) N leaching rate shares of applied mineral N fertilizer were decreased faster and more

strongly than in NT compared to in REF over the course of the simulation period (Fig. 2 d). After the first three initial years the N leaching rate responses were stabilizing for all three alternative scenarios.

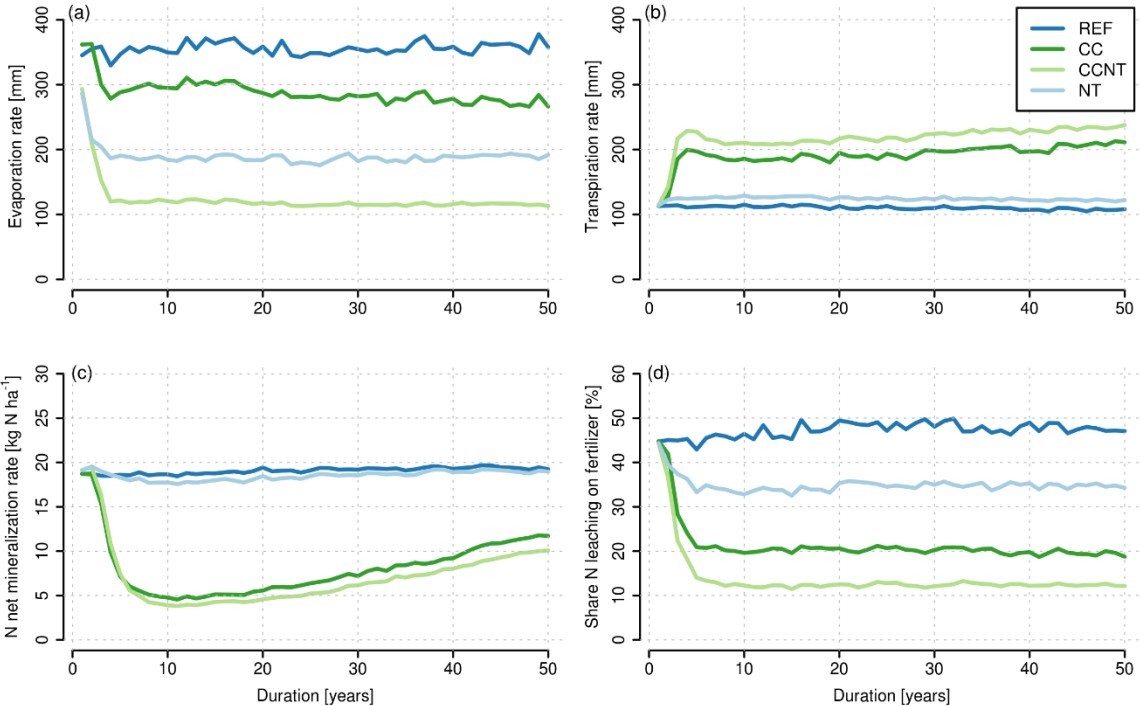

**Figure 2** Plots in panel display the time series for the 50 year simulation period of the annual global spatial aggregated area-weighted median per hectare cropland (~1,500 million ha) as lines per management scenario (REF: dark blue, CC: dark green, CCNT: light green, and NT: light blue) for: (a) Evaporation rate in mm, (b) Transpiration rate in mm, (c) Soil N net mineralization rate in kg N ha$^{-1}$ (derived as absolute difference between soil gross N mineralization and immobilization rates), and (d) Shares of annual soil N loss through leaching of applied mineral N fertilizer rate in percent (%).

The relative differences in soil N leaching rates compared to the baseline (REF) are illustrated in Fig. 3 and indicate a reduction on the majority of global cropland in all three alternative soil management scenarios (for the respective spatial pattern of changes obtained for the cover crop scenario (CC) see Fig. S2.1.2). Larger reductions and lower spatial variation are generally found during the last than during the first decades of the 50 year simulation period. Median reductions in N leaching rates in simulations including cover crops (CC and CCNT) were about two to three times higher than in NT.

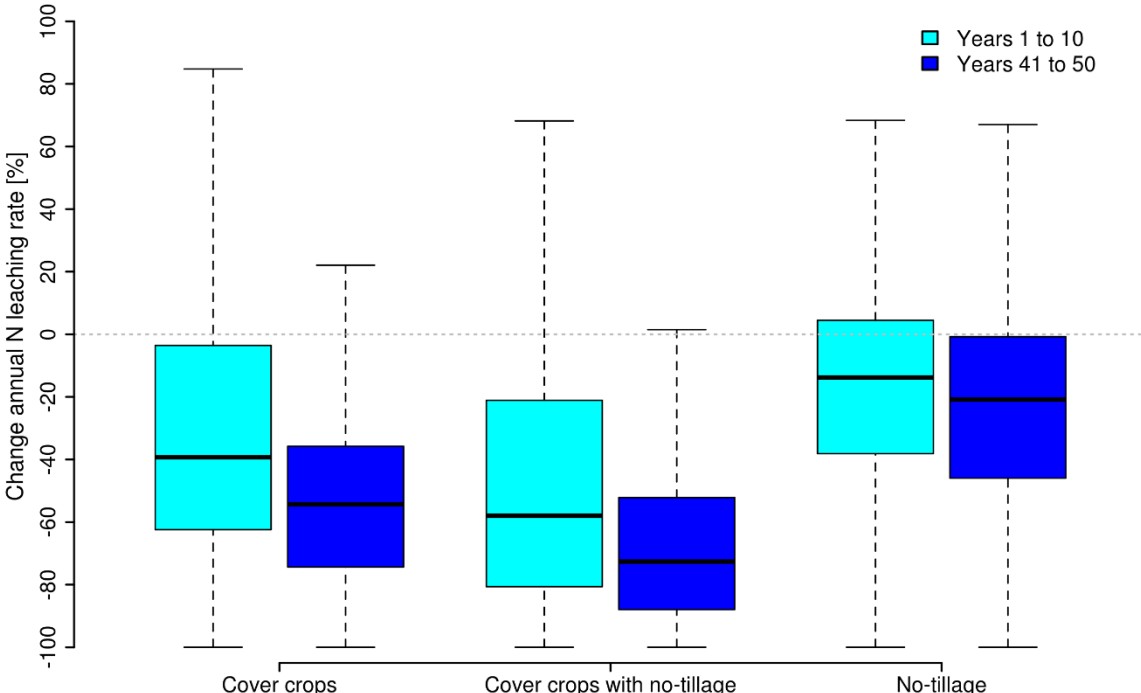

**Figure 3** Boxplots of relative differences (%) per hectare cropland (~1,500 million ha) between annual N leaching rates in each of the simulated alternative management scenarios (CC, CCNT, and NT) compared to the baseline (REF) in the first (left bars, cyan) and last decades (right bars, blue) of the 50 year simulation period. The black midlines of boxes indicate the median responses per period, hinges of boxes show the first (Q1) and third (Q3) quartiles, and whiskers extend both to the minimum and maximum values within 1.5 times the interquartile range (IQR) of the distribution (outliers, defined as values outside this range are not shown here).

### 3.3 Yield change of following main crop due to altered management and duration

The simulated impacts of cover crop cultivation (CC) on the following main crop yields exhibited large spatial variability and differences of effects between the analyzed crop types. The productivity for maize and rice in northern cold and tropical humid climates was lowered with cover crops (CC), whereas drier temperate regions e.g., in the Western USA and Mediterranean reveal prominently enhanced average yield effects for the four assessed crop types (Fig. 4).

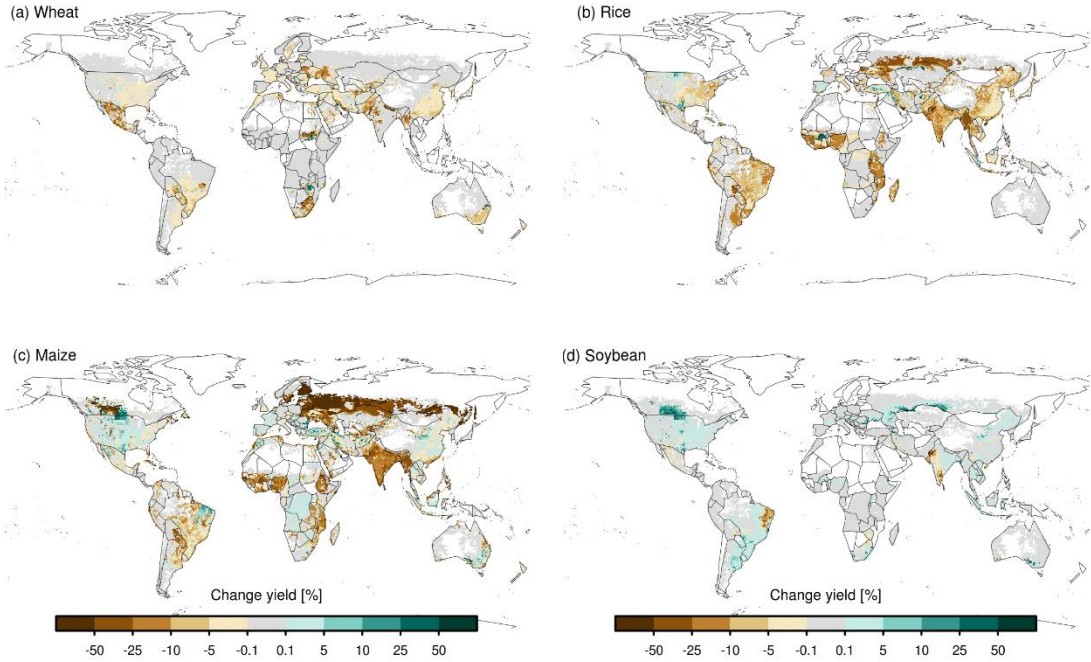

**Figure 4** Maps (a-d) showing changes of averaged rainfed and irrigated main crop productivity in response to cover crops (CC) compared to the scenario with bare fallow on cropland during main crop off-season periods (REF) as annual median relative differences in percent (%) per hectare of crop-specific cropland and grid cell (pattern of the year 2010, Sect. 2.3) for wheat, rice, maize, and soybean for the 50 year simulation period.

Comparing the changes across the alternative management scenarios, following main crop average productivity decreased most strongly in CC and increased most in NT relative to the baseline with tillage and bare soil fallow practices (REF) (Fig. 5 a-d). In CC, rice yield declines were largest, whereas reduction for this crop type was halved on the majority of global cropland in the CCNT simulation. In contrast to mostly lowered maize yield in CC, we found positive median responses for this crop type in CCNT but with higher spatial variability of impact magnitude and direction (Fig. 6, Table S2.2). Wheat yield responses to any of the three alternative managements were very low in overall magnitude, being slightly reduced in both cover crop scenarios, but improved in NT (Fig. 5, Table S2.2). Soybean yield, responded positively to all simulated alternative management practices, although by median less than 1 % in CC, we calculated around 9 % higher medians in CCNT, and NT compared to in REF. Exploring simulated land management impacts on the following main crop productivity separated by water regimes revealed larger spatial variability of responses for rainfed than for irrigated crop yields (Fig. S2.4). Soybeans in irrigated systems show no response to altered management practices. For the other assessed cereal crop types', median yield responses to cover crop practices (CC, CCNT) were found to be either more negative or changing from a positive to a negative response in irrigated systems compared to rainfed systems.

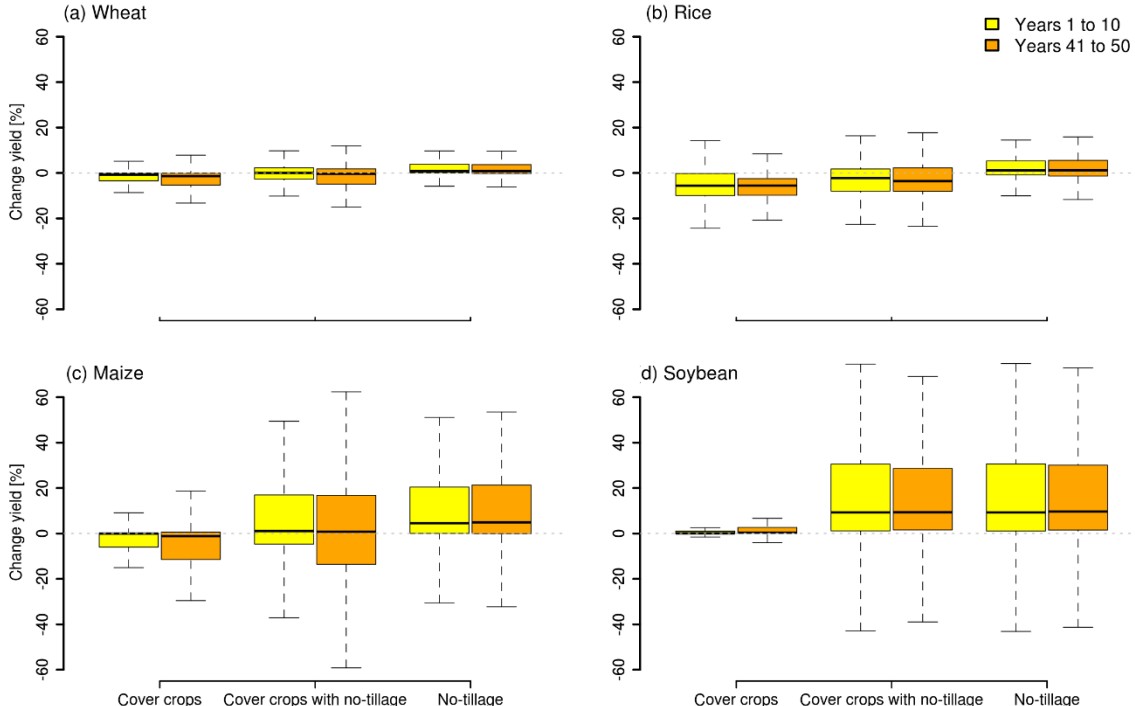

**Figure 5** Panels (a-d) displaying changes of wheat, rice, maize, and soybean average yields as boxplots of relative differences in percent (%) area-weighted by crop-specific physical cropland, due to alternative management practices (CC, CCNT, and NT) compared to the baseline scenario (REF) for the first (left bars, yellow) and last decades (right bars, orange) of the 50 year simulation period. Boxes' black midlines indicate the spatial median across the distribution of responses, the lower and upper edges of the boxes the first and third quartiles, and whiskers extending both to the minimum and maximum values within 1.5 times the interquartile range, respectively from each Q1 and Q3 (outliers, defined as values outside this range are not shown here). The boxplots show the distribution of calculated responses across total crop-specific irrigated and rainfed physical cropland used for the year 2010 for wheat (~333 million ha), maize (~369 million ha), rice (~132 million ha), and soybean (~94 million ha). Irrigated shares of total global crop-specific physical cropland were 16 % for wheat, 12 % for maize, 35 %  for rice, and 11 % for soybean cropping system area (Sect. 2.3).

### 3.4 Cover crop and no-tillage impacts on Conservation Agriculture cropland

Applying the obtained responses to altered land management practices to the temporal and spatial pattern of the mapped CA cropland time series dataset (Sect. S1.4), we found lowest soil C sequestration rates and reductions of N leaching rates assuming no-tillage practices and highest for combined cover crop and no-tillage practices (Table S2.5). We calculated aggregated median soil C sequestration rates of 0.27 t C ha$^{-1}$ yr$^{-1}$ for no-tillage and bare soil fallowing, 0.47 t C ha$^{-1}$ yr$^{-1}$ for cover crops with tillage, and 0.85 t C ha$^{-1}$ yr$^{-1}$ for cover crops with no-tillage. We estimated the total historical soil C net-accumulation by CA practices on the mapped cropland ranging between 0.4 - 1.4 PgC in the period 1974-2010, depending on the management practice. For the annual N leaching rates, we find the reduction by the single or by the combined adoption of no-tillage and cover crop practices ranging between 18.4 - 56.9 %  across global CA area compared to cropping systems with conventional tillage and bare soil fallowing practices.

We found average yields of the four main crops mostly enhanced with no-tillage, whereas for cover crop with tillage practices the productivity response was neutral or revealed decreases. In response to cover crops applied with no-tillage practices (CCNT), which scenario we used as proxy for the full set of CA practices, positive yield changes (Fig. 6) dominate in areas mapped with Conservation Agriculture practices (Fig. S1.4).

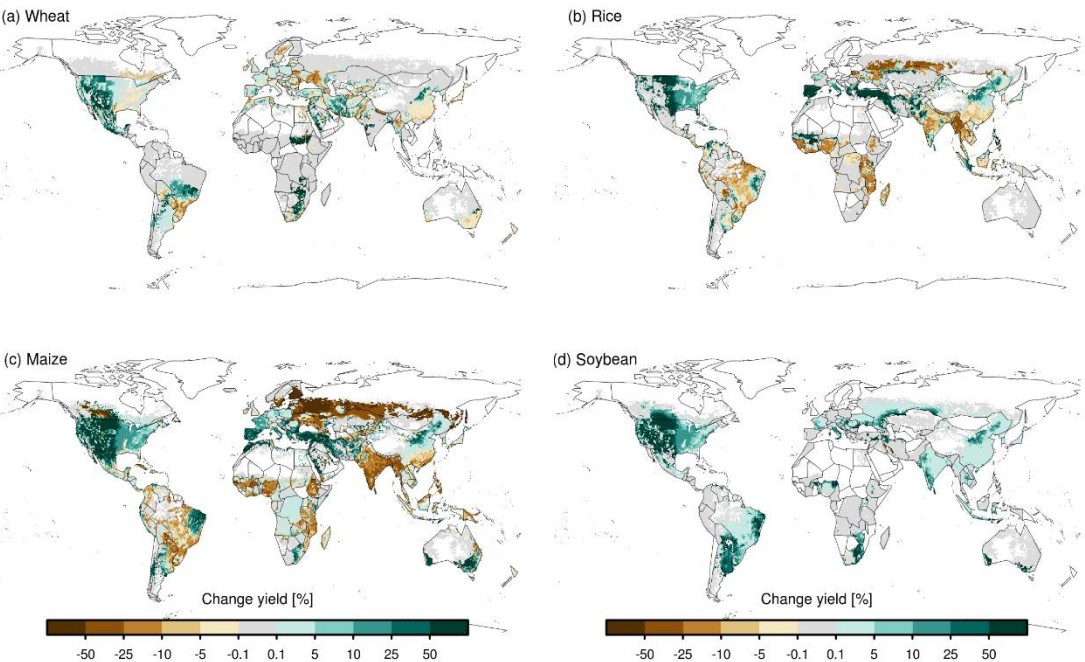

**Figure 6** Maps (a-d) showing changes of main crop average productivity in response to cover crop practices combined with no-tillage (CCNT) compared to the baseline scenario with conventional tillage and bare fallow on cropland during main crop off-season periods (REF) as annual area-weighted median relative differences in percent (%) of crop-specific cropland and grid cell (pattern of the year 2010, Sect. 2.3) for wheat, rice, maize, and soybean for the 50 year simulation period.

Calculating median (quartiles) for yield changes on CA areas only, we found for cover crop with no-tillage that the average productivity (median (quartiles)) of wheat, maize, and soybean was almost exclusively enhanced (6.4 (0.2, 29.4); 23.7 (3.3, 84.1); 27.8 (3.1, 79.0) %, respectively). Although rice yield largely increased with the combined practices but can be lowered as well (5.6 (-3.1, 34.8) %).

## 4 Discussion

### 4.1 Overview of simulated responses to cover crop practices compared to other studies' findings

Simulated cover crop impacts exhibit positive soil carbon sequestration rates and reduced N leaching rates on the majority of global cropland, but at the cost of largely lowered average yields of the following main crops in both analyzed decades (Table 1 see Table S2.2 for respective aggregated results per decade for CCNT and NT) The here estimated changes of agroecosystem components due to cover crops (CC) compared to bare soil fallow (REF) on cropland between two consecutives main crop growing seasons, are consistent with the magnitude and direction of effects reported in other studies (Table 1, see Supplement Table S2.6 for an extended comparison to literature values,.

**Table 1** Simulated responses to cover crops (CC) in comparison to the control scenario with bare fallow (REF) on cropland during main crop off-season periods as annual aggregated area-weighted median and in the parenthesis the quartiles (Q1, Q3) for the first and last decades of the 50 year simulation period, respectively, (see Sect. 2.5 for equations used). In the latter two columns values from other studies as well as their considered duration of cover crop management are reported.

| | Unit per year | Simulated ΔCC first decade median (quartiles) | Simulated ΔCC last decade median (quartiles) | Literature ΔCC range of values (min.-max.) | Management duration (years) |
|---|---|---|---|---|---|
| **Soil C sequestration rate** | t C ha$^{-1}$ | 0.52 (0.03, 1.04) | 0.48 (0.24, 0.78) | 0.01 - 0.56[a] | 1 - 54 |
| **N leaching rate** | % | -39.3 (-64.2, -3.6) | -54.3 (-74.4, -35.8) | -70 - (-50)[b] | 1 - 17 |
| **Wheat yield** | % | -0.7 (-3.5, 0) | -1.4 (-5.3, -0.1) | | |
| **Rice yield** | % | -5.6 (-9.9, -0.3) | -5.6 (-9.8, -2.5) | | |
| **Maize yield** | % | 0 (-6.0, 0.1) | -1.2 (-11.5, 0.6) | 0 - 9.6[c] | 5 |
| **Soybean yield** | % | 0.1 (0, 1.0) | 0.4 (0, 2.7) | 2.8 - 11.6[d] | 5 |
| **Average change in yield** | % | -1.6 | -2.0 | -4 - 0[e] | 1 - 28 |

[a] Jian et al. (2020); Lal (2004b); Paulsen (2020); Poeplau and Don (2015); Sommer and Bossio (2014); Stockmann et al. (2013)

[b] Thapa et al. (2018); Tonitto et al. (2006); Valkama et al. (2015)

[c] Marcillo and Miguez (2017); SARE (2019)

[d] SARE (2019)

[e] Abdalla et al. (2019); Thapa et al. (2018); Tonitto et al. (2006); Valkama et al. (2015)

**4.2 Soil carbon sequestration**

The generated median soil C sequestration rates for the simulation with cover crops replacing bare soil fallow periods were within the upper end of range of values reported in the literature (Table 1, Table S2.6). Few regions in temperate and dry climatic conditions, e.g. in Western USA, Turkey, Iraq, and Iran, reveal a neutral or declining

trend in response to cover crop cultivation (Fig. S2.1.1). In line with findings of West and Six (2007), we found highest soil C sequestration potential in tropical regions (e.g. South-East Asia and Central Western Brazil), whereas Stockmann et al. (2013) derive largest potential for temperate humid regions. Abdalla et al. (2019) find both regions profiting from cover crop practices, because there, water is a less limiting factor to biomass production and additional inputs to the soil pools provided by cover crops, enhance soil C accumulation.

Assuming the median soil C sequestration rate of 0.55 t C ha$^{-1}$ yr$^{-1}$ (or mean of 0.61 t C ha$^{-1}$ yr$^{-1}$) during a period of 50 years for the estimated 400 million hectare cropland potentially available annually for cover crop practices (Kaye and Quemada, 2017; Poeplau and Don, 2015), we estimated the global potential soil C sequestration of 0.22 or 0.24 PgC yr$^{-1}$ in the top 30 cm. This equivalents to about 7-12 % of the 2-3 PgC yr$^{-1}$ annual sequestration on global agricultural soils until the year 2030 targeted by the '4per1000' initiative (Minasny et al., 2017). However,

our estimate is higher than the 0.12 ± 0.03 PgC yr$^{-1}$ (mean and standard deviation) found by Poeplau and Don (2015) simulating cover crops effects with the RothC model for a similar time frame but for 0 - 22 cm soil depth. Lower annual median soil C sequestration rates with cover crops (CC) in the first three simulation years, reveal a time lag of response to altered management (Fig. 1). A similar effect is also apparent for N and water fluxes (Fig. 2). On the one hand, this may be because cover crops are first established at the end of the first main crop growing season, so that the full effect becomes visible in the second year only. On the other hand, a temporal delay of detectable cover crop impacts on soil organic C concentration within the first years of practice was also found in the review of ecosystem services of cover crop practices by Blanco-Canqui et al. (2015), due to the complexity of biophysical processes affected by changes in biomass inputs to soils due to altered management practices. This suggests that cover crops need to be cultivated for at least three years to take effect. Duration, as the number of years a system has been under a management practice, was also identified as one of the most important factors to reap the benefits of altered soil physical properties from soil C storage enhancing management, such as cover crops practices (Laborde et al., 2020; Nouri et al., 2020; West and Six, 2007).

The higher soil C sequestration rates calculated for the first than for the last decade of the 50 year simulation period (Table 1, Fig. 1) are in line with other studies' estimates as well. For example Sommer and Bossio (2014), assumed their soil C sequestration rate functions for their simulations of cover crop impacts to peak between the third and seventh year of continuous practice and then to level off after about 20 to 40 years. Corsi et al. (2012) in their meta-analysis on effects of CA practices, found a decreasing rate of soil C sequestration between the fifth and twentieth years. The decreased change rates towards the end of the 50 year simulation period, suggest a saturation effect (for cover crops later than for no-tillage), when soil C and N pools approach a new equilibrium state, as discussed by Kaye and Quemada (2017); Poeplau and Don (2015); Smith (2016). However, the new equilibrium of soil C (Corbeels et al., 2018; Poeplau and Don, 2015) were not reached in our simulations for the majority of global cropland for CC or CCNT within the analyzed 50 years. For NT, half of global cropland reached the new equilibrium after 12 years. Our assumption on 'equilibrium' as effect detected in our alternative management simulations on global cropland was based on Poeplau and Don (2015), who define the new steady state to be reached after the annual change in the soil organic C stock fell below 0.01 Mg C ha$^{-1}$ yr$^{-1}$ in response to altered management.

The soil carbon sequestration effect of no-tillage practices simulated with LPJml5.0-tillage have been evaluated in Lutz et al. (2019) who used a preceding model code version to the one employed here but used another simulation setup and different main crop residue management settings. They calculated a median soil C stock increase of 5.3 % (after 10 years) for their stylized no-tillage scenario with retaining all main crop residues on the field after harvest, and -18 % (after 20 years) in their other management no-tillage scenario with 90 % main crop residue removal rates. However, in our simulation main crop residue removal rates vary across global gridded cropland (see Sect. 2.3) and therefore the modeled results by Lutz et al. (2019) can only partly be compared to our values (relative differences of 2 % for CCNT to CC, and 1.3 % for NT to REF).

The median soil C sequestration rate for both cover crop scenarios (CC and CCNT) were higher than for no-tillage (NT) (Fig. 1, Table 1, Table S2.6), which is in line with findings of the review by Kaye and Quemada (2017). The effect of combined cover crop and no-tillage practices (CCNT) exhibited the largest soil C sequestration rate with median 0.72 t C ha$^{-1}$ yr$^{-1}$ in the first decade. Our result were higher than Franzluebbers (2010) finding a soil C sequestration rate of 0.45 ± 0.04 t C ha$^{-1}$ yr$^{-1}$ for experiments comparing cover crops combined with tillage and

no-tillage in Southeast USA for about 11 years and were within the range stated in the meta-analysis of experiments conducted in Brazil (0.4-1.9 t C ha$^{-1}$ yr$^{-1}$) and France (0.1-0.4 t C ha$^{-1}$ yr$^{-1}$) with a duration of 5-28 years by Scopel et al. (2013). The simulated higher effect of cover crops combined with no-tillage (CCNT) on soil C stocks is also supported by Corbeels et al. (2018) finding higher soil C stocks in case of CA compared to conventionally tilled systems, whereas Abdalla et al. (2019) and Poeplau and Don (2015), find no significant differences due to changed

tillage practices with cover crops in their meta-analyses.

### 4.3 Nitrogen leaching

The derived N leaching rate reduction in CC were at the upper end of the -70 to -50 % range of effects reported in literature (Table 1, Table S2.6), except during the initial simulation years after introduction of the alternative management practice. The majority of values found for changes due to cover crops reported in other studies and

used here for comparison to our obtained simulation results, are for rainfed cropping systems or detailed management information on irrigation practices was missing (Table S2.6). However, Quemada et al. (2013), explicitly focusing in their meta-analysis on cover crop effects replacing bare fallows in irrigated cropping systems, also state a reduction of N leaching rate by 50 % with non-leguminous cover crop species but no effects in experiments with leguminous cover crop types.

For the spatial effects of cover crops (CC), it can be depicted, that most cropland can profit from about halved N leaching rates (Fig. 3, Fig. S2.1.2). One important driver of the size of the effect of cover crops is the length of the fallow season. In northern regions, main crop growing seasons are rather short and aligned across crop types, so that a lot of off-season cropland area is available for cover crops for relatively longer time. Largest N leaching rate reduction with simulated cover crop practices can be found in cold temperate regions (such as in Russia) and in

humid tropics (e.g., large parts in Africa), where external N inputs (i.e. mineral N fertilizer rates, also see Sect. S1.2 for rates used here) are rather low. On the one hand, the variance of simulated cover crop effects across global cropland can be attributed to management intensity (e.g., fertilizer application rates), in this study prominently seen as differences at some national borders (USA and Canada). According to Wittwer et al. (2017) efficiency of cover crops to reduce N leaching is decreasing with management intensity (including fertilizer application rates

and tillage practices). On the other hand, the spatial variance of cover crop effects within countries suggest differences due to soil and climatic conditions. Only few drier regions reveal either a neutral response or slight increase of N leaching rates due to cover crops (Fig. S2.1.2). This can be attributed to reduced growth of cover crops, limiting their capacity for N uptake of excess N remaining in the soil column after harvest of the main crop. Cover crops were also found to increase transpiration while reducing drainage (Meyer et al., 2018), which leads

to lower soil water percolation (Abdalla et al., 2019) and restrict the advective export of reactive N from the soil. However, in dry regions, depending on the area share and the type of irrigation system, the additional water applied to fields can result in enhanced drainage. The effect is pronounced for surface irrigation and weaker for drip irrigation. As a result, for irrigated cropping systems in dry areas, N leaching may increase with cover crop practices due to increased biomass inputs to soil which lead to increased N in the soil water solution as a result of

decomposition processes of the added plant material.
Because the plant material from cover crops that drives the C sequestration with the practices (Sect. 3.1, 4.2) has a wider C to N ratio than the soils, it leads to stronger immobilization of mineral N in the soil column (Fig. S2.3). Increased evapotranspiration and immobilization but also uptake of N by cover crop plants were found to reduce

the soil N (Quemada et al., 2013; Thapa et al., 2018; Zhu et al., 2012), which would be susceptible to leaching
from cropland soils during primary crop off-season periods (Abdalla et al., 2019; Alonso-Ayuso et al., 2014; Delgado et al., 2007; Tonitto et al., 2006). For their efficiency in N uptake, grass cover crops are also described as 'scavengers' (Blanco-Canqui et al., 2015). Therefore, grass cover crops can be regarded especially suitable for high-input farming systems, where surplus N left in the soil after harvest of the main crop can be retained in the biomass of the cover crop. After termination, the C and N contained in the cover crops biomass, can serve as 'green
manure' temporally fixed in compounds of the soil organic matter (Zomer et al., 2017).

### 4.4 Crop yields

The average main crop yield change computed for the cover crop scenario (CC) were mostly within the range of values found in literature, but effects vary largely per crop type and location considered (Table 1, Table S2.6). Reduced productivity levels of the following main crop are reported mostly in the context of competition with the
cover crops for water and nutrients (Abdalla et al., 2019; Tonitto et al., 2006; Valkama et al., 2015). The increased immobilization of soil N after the introduction of cover crops is thought to actually exacerbate N stress (Abdalla et al., 2019; Erenstein, 2003; Kuo and Sainju, 1998; Ranaivoson et al., 2019). Marcillo and Miguez (2017) assume that lower maize yields found with cover crops may also be caused by a temporal asynchrony between periods of soil N mineralization and high N demand of the main crop. Several authors (Marcillo and Miguez, 2017; Thapa et
al., 2018; Tonitto et al., 2006) report no significant effects of non-leguminous cover crop species on yields of the subsequent main crop, which may be caused by the mainly intensively fertilized experiments considered, e.g. in Tonitto et al. (2006). This is in line with our findings for soybean, which is an N fixer (not subject to N limitations in LPJmL) and sees hardly any yield penalty from cover crops. Also, the mostly negative responses to cover crops (CC) for the three cereal crop types in irrigated systems (Fig. S2.4.2), where water is not a growth-limiting factor
for the main crop, can only be explained by a decrease in N availability for the main crop. In the meta-analysis by Quemada et al. (2013) a reduction of irrigated main crop yields by 3 % was found due to cover crops, which effect is slightly higher than the decadal median reductions of the following main crop yields by 2.5 % and 2.9 % (average across changes of irrigated wheat, rice, maize, and soybean yields for the first and last decade of the 50 year simulation period). The generated results for irrigated soybean productivity reveal no sensitivity to cover crop and
changes in tillage practices (Fig. S2.4.2). In our simulations the majority of cropland of wheat maize, rice, and soybean was rainfed (see caption of Fig. 5). Therefore, the found neutral or positive responses of following main crop average yield to cover crop practices for temperate dry areas in the US and the Mediterranean region may result from the relatively higher mineral N fertilizer application rates there (Fig. S1.2), and larger shares of irrigated cropping system area on cropland per grid cell, wherein the effects of cover crops' competition for water and
nutrients with the following main crop are diminished. Cover crops affect soil water in different ways: cover crops tend to increase transpiration (see Fig. 2 b), but at the same time reduce soil evaporation (Fig. 2 a) and increase infiltration (Dabney et al., 2001). Depending on the relative magnitude of these processes, soil water availability for the main crop can increase or decrease at different locations. This is clearly shown in Fig. S2.4, where yield responses to cover crops in rainfed systems reveal a much larger variability than in irrigated systems. The spatial
variability of yield responses to cover crops for different crops (Fig. 4 and 5) is the result of differences in how cover crops impact water availability of the main crop, how water limited the main crop is, and how strongly the cover crop the reduces N availability for the main crop. However, sensitivity to changes in water availability is

highest in rainfed systems in water limited environments, on soil types of low soil water holding capacity, or insufficient recharge, which limits their applicability under such conditions (Marcillo and Miguez, 2017).

In contrast to CC, a mostly enhancing effect on productivity was found with the NT scenario for all four analyzed main crop types. Also Su et al. (2021) find for wheat, maize, and soybean, that although no-tillage could lead to yield declines in cooler and wetter regions, this loss to be more than compensated at the global scale by increased productivity in arid rainfed cropping areas. In our model, the yield increase can mainly be attributed to the water-saving effects simulated with no-tillage compared to both REF and CC scenarios with conventional tillage (Fig. 2, Fig. 5). This is caused by the built up of a litter layer due to simulated no-tillage practices covering the soil as mulch, which increases infiltration rates as well as reduces evaporation and surface runoff rates, mainly benefitting soil water dynamics and crop productivity in arid regions (Jägermeyr et al., 2016; Lutz et al., 2019). Lutz et al. (2019) estimated the difference between tillage to no-tillage for rainfed yields of wheat of median 2.5 % in the simulation with all main crop residues retained, and -5.9 % with 90 % of the residues extracted from the field. For rainfed maize yields they found 1.8 % median increases in their simulation with all main crop residues retained, and -5 % when 90 % residues extracted, after 10 years since the introduction of no-tillage practices. Our calculated changes in yields due to no-tillage are within these ranges (rainfed wheat: 1.7 % for CCNT to CC, and 1.3 %  NT to REF; rainfed maize: 4.8 % for CCNT to CC, and 6.3 %  for NT to REF as aggregated relative differences for the simulated years 9-11).

In CCNT, the simulated effects of cover cops and no-tillage are combined. Cover crops provide vegetative soil cover on cropland during main crop off-season, and when terminated serve as additional mulching material during the following main crop growing periods. This additional mulch layer in combination with no-tillage counteract the higher transpiration from cover crops by improving infiltration and reducing evaporation (Abdalla et al., 2019; Scopel et al., 2013). Enhanced median maize and soybean yields, as well as less rice yield reductions found with CCNT than with CC compared to REF (Table 1, Table S2.4), reveal co-benefits of both practices (Fig. 5). The assumption of synergetic effects of both practices in CCNT were supported by the even higher median yield responses derived here for cropland with Conservation Agriculture practices (Sect 3.4), which area was mapped with a higher likelihood to arid regions (Porwollik et al., 2019). Laborde et al. (2020) find higher likelihoods of beneficial main crop yield effects of CA for rainfed cropping systems in areas with higher temperature (above 20 degree Celsius) and lower precipitation rates (< 350 mm), due to water-preservation, when the mulching practices reduce evaporation losses compared to experiments with conventional land management practices.

The here presented yield responses to different management settings (NT, CCNT) are only partly in line with findings of Pittelkow et al. (2015), analyzing experiments lasting 1-31 years, who find largest declines (-9.9 %) when no-tillage was adopted alone and decreased negative effects (-6.2 %) when no-tillage was applied with crop rotation. However, cover crops as modelled in our CCNT scenario are only one aspect of crop rotation enhancement considered in the analyses by Pittelkow et al. (2015), which limits the comparability between our and their findings.

### 4.5 Methodological limitations and implications

A detailed evaluation of the implemented model code functionalities for the representation of cover crop practices remains challenging, due to the limited available statistical data on the practice at the global scale. We mainly focused the comparison of modeled effects to findings of meta-analyses and reviews. The results indicate in general

reliability of the here used model version LPJml5.0-tillag-cc to reproduce ranges of reported temporal and spatial pattern, magnitude, as well as the sign of direction of simulated cover crop impacts at the global scale (Table 1). However, aggregated changes of agroecosystem variables due to cover crop cultivation (CC) compared to bare fallowing practices presented here, were not always matching other studies' findings (Table S2.6). On the one hand, these deviations may result from different soil depth considered or meta-analyses reporting averages across different years, crop types, and experiments (Nyawira et al., 2016). Further uncertainties are related to literature values, which may include experiment results from measurements during the main crop growing season only instead of covering the entire year (Quemada et al., 2013). Values derived from field experiments or reported at the national scale (Table S2.6) may rather reflect changes due to local specific and highly controlled crop production conditions rather than covering the variance of environmental and socio-economic conditions captured with the global gridded model setup applied in our analysis. On the other hand, important processes that determine the effect of cover crops in field trials, such as erosion, weeds, pests, or diseases, are not accounted for in this model version.

For the 50 year simulation period we used dynamic historical climate model input data for the years 1962-2011 and all stylized management scenarios were introduced from the year 1962 onwards. Therefore, the first decade after the introduction of the practice is not directly comparable to the last decade in terms of environmental cropping system conditions but we assume this error to be small, given that we average over larger areas and report differences to values obtained in the baseline scenario that always refers to the same simulated period when reporting our results.

Further, our simulations include changes in atmospheric $CO_2$ concentration levels during the spin up and historical simulation period, which affect soil C dynamics as well., through biomass growth feedbacks but also temperature and soil moisture effects driving decomposition of the soil organic carbon in cropland soils (Herzfeld et al., 2021). However, some of the field experiments to which we compare our simulated management results have been conducted over comparable time periods (up to 54 years) and are therefore affected by increasing atmospheric $CO_2$ concentration levels during the near-past period, as well.

The high C sequestration rates calculated for CC, e.g. in the humid tropics (Fig. S2.1.1) may be due to an overestimation of the simulated fallow period length for cropland in this climatic region. In the model version used here, only the main representative growing season of a crop is simulated per year, so that multiple cropping practices for areas where several crop harvests per year are common (Siebert et al., 2010; Waha et al., 2020) are not well covered, resulting in distorted cover crop productivity levels and biomass input to the soil pools.

The here applied model setting for the representation of irrigated cropland in the simulations, assuming unlimited water availability for irrigation practices, may cause an overestimation of main crop productivity as well as resulting main crop residue input amounts to the soil pools.

The computed initial soil C pools do not represent the conditions on current croplands because our simulations excluded historical land use dynamics, to which responses in soil usually are slow and of long-term (Nyawira et al., 2016). Pugh et al. (2015) find, that the soil legacy flux from land use and land cover change may dominate ecosystem carbon losses for a timescale up to a century. By starting the simulations from soil C pools in equilibrium, we aimed to make sure that the acquired response is due to altered management. The deviations in initial soil C and N pools were accounted for in this study by presenting responses to alternative management scenarios (CC, CCNT, NT) in relation to the baseline scenario (REF).

The simulated crop-specific yield reductions and gains in soil carbon storage obtained at the grid scale for the cover crop management scenarios (CC and CCNT) as a trade-off cannot always be linked directly, due to the missing accounting of emissions associated to the changes in management in our study. Further, modeled responses to cover crop cultivation are determined by the spatial pattern of the crop type, the area share within a grid cell, the crop specific growing season length, fertilizer application rates (Fig S1.2b), the water regime (S2.4), and other crop management modeled at the grid scale.

The potential trade-off between environmental benefits (reduced N leaching, soil C sequestration) and main crop productivity changes found here for cover crops with conventional tillage practices, suggest the requirement for the complementary modification of fertilizer management and of main crop irrigation or the parallel adoption soil water preserving practices, such as no-tillage, and mulching practices to maintain current main crop yield levels. Our findings on nearly neutral effects on the four analyzed following main crop yields in irrigated cropping systems with cover crops (CC) may result of the here used potential irrigation setting in the management scenarios-providing unlimited water amounts to satisfy completely main crop plant growth requirements during the entire growing season. The obtained results may underestimate yield declining effects because local specific limitations to irrigation water withdrawal amounts are not accounted for in our analysis. However, it can be assumed that increased irrigation water requirements, because of the increased evapotranspiration losses from cover crops, may constrain main crop yield gains obtained with irrigation when adopting the practice with conventional tillage in dry areas.

This study is the first to consider combined cover crop and no-tillage effects as practices recommended under CA employing modeled results employing the generated annual gridded CA area time series dataset.. However, we assume that the here employed method for mapping CA area led to several uncertainties of estimated effects on agroecosystem components. The downscaling approach of national reported CA area to the grid scale targeted a coarser resolution than used in Porwollik et al. (2019) as well as included the entire cropland within a grid cell assumed under this practice, independent of the crop type and water regime of the cropland, whereas in the previous approach only rainfed cropping system area was considered as suitable. The national CA area statistics reported in the FAO also includes area of perennial crop types, which in LPJmL are represented as the CFT 'others' with annual growth dynamics only, and of grassland, which dynamics are not included in our analysis.

Further, our management simulation scenarios do not include a change in main crop residue removal rates, which historical rates may deviate from minimum levels of 30 % soil surface cover by biomass remaining on the field after harvest required by CA. Also, secondary usages of cropland products are likely, so that our estimates of cover crop biomass input rates on the mapped historical CA area maybe overestimating, when assuming all cover crop biomass remaining on the field in the CC and CCNT scenario. Nevertheless, the dataset was used as model input data for simulating historical tillage and no-tillage practices in Herzfeld et al. (2021) to assess global soil C dynamics. The 'partial adoption' of CA practices, which mostly refers to the adoption reduced tillage but not necessarily to the diversification of the crop rotation and the soil cover management as suggested under CA, was discussed as uncertainty related to reported CA area included in the national FAO statistics (Porwollik et al., 2019; Prestele et al., 2018). Therefore, the uncertainty of the historical ecosystem services provided by historical CA cropping systems due to reporting schemes in the literature and statistics, as well as the here used model, mapping, and calculation approaches may be better reflected by the range of values obtained for the three alternative management practices assessed here (Table S2.5). Obtained results exhibit large variations across space, time, and

management scenarios, so that upscaling efforts of the practice need to account for differences in environmental and socio-economic conditions of cropping systems.

Further global scale modeling assessments of sustainable land management practices may include leguminous (N fixing) cover crop species, or mixes of them with the here presented grass type. Production costs associated to additional irrigation water requirement and seed purchase for cover cropping (Alonso-Ayuso et al., 2020). Opportunity costs for field activities of the farmer in otherwise off-season periods (Lee and Thierfelder, 2017) need to be evaluated in integrated assessments against the environmental benefits from cover crop practices (Blanco-Canqui et al., 2015). Further studies are needed for the quantification of cover crop impacts with climate change and to explore options for adaptation of the practice to regionally specific environmental and economic conditions, influencing farming decisions and land management practices.

## 5    Conclusion

This study presents the first global temporal and spatially explicit quantification of impacts of cover crop cultivation in combination with tillage practices. The routines of cover crops implemented into LPJmL5.0-tillage-cc, allow for consistent, global-scale assessments of biophysical, biogeochemical, and agronomic effects, such as on mapped CA cropland during the period 1974 to 2010 and for exploring potentials of sustainable cropland management practices.

We found, that cover crops enable soil C sequestration and reduce N losses through leaching on the majority of global cropland, except in few and mostly unproductive arid regions. Cover crop with conventional tillage practices increase evapotranspiration fluxes and decrease soil N net-mineralization rates compared to bare soil fallowing practices by lowering plant available soil water and nitrogen, leading to reduced growth and yield of the following main crop. Declining average yield effects due to cover crops were found for rice, but also for maize, and wheat, most pronounced for cropping areas in northern cold climatic regions. Enhanced productivities with cover crops and tillage for these three staple crops were depicted for temperate regions with high mineral N fertilizer application rates and for almost all soybean production.

The yield responses to altered management generated for all four crop types were rather constant over time, whereas for changes in soil N leaching rate and C sequestration pronounced temporal dynamics were found. For soil C sequestration and N leaching the sign of changes was mostly homogeneous across global cropland, whereas for productivity, the direction and magnitude of changes vary considerably among crop types and for different world regions.

For cover crops applied with no-tillage (CCNT), both the soil C sequestration rate and the reduction of N leaching were largest. The combined practices take advantage of the additional biomass production by cover crops and of the soil water saving effects associated to no-tillage, which results in increasing inputs to the soil, improved nutrient cycling, and substantially reduced rainfed crop yield penalties.

We conclude from the findings, that the heterogeneity of cover crop impacts on C, N, and water processes are determined by the primary crop type cultivated, water regime (rainfed or irrigated), tillage and mulching practices, location, as well as management duration. This study's results demonstrate the potential role of cover crop practices as a nature based solution (Keestra et al., 2018) to transform croplands to C sinks for climate change mitigation and the reduction of environmental impacts of arable production without compromising food security targets.

*Code and data availability.* The LPJml5.0-tillage-cc model code version, model output data, and R-scripts used for post-processing data accompanying this study are available online at the Zenodo data repository: https://doi.org/10.5281/zenodo.5178070 (Porwollik et al., 2021).

*Supplement link.* The supplement related to this article is available online at:

*Author contributions.* VP, CM, SR, and JH designed the research. VP and CM implemented the cover crops code functionalities with the support of all other authors. VP generated the CA cropland dataset and conducted the simulations. VP and CM analyzed results. VP prepared the manuscript and all co-authors contributed by commenting and editing.

*Competing interests.* The authors declare that they have no conflict of interest.

*Acknowledgements.* We thank Tobias Herzfeld for support in model code development, Kristine Karstens for constructive discussions on cropland soil carbon, as well as Jan Kowalewski and Jannes Breier for data processing contributions.

*Financial Support.* SR and VP acknowledge financial support throughout the MACMIT (01LN1317A), SR also from the CLIMASTEPPE (01DJ18012) and JH from the EXIMO (01LP1903D) projects, all funded through the
735 German Federal Ministry of Education and Research (BMBF).

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
