# Peer review of "The role of cover crops for cropland soil carbon, nitrogen leaching, and agricultural yields - A global simulation study with LPJmL (V. 5.0-tillage-cc)"

_Biogeosciences, 2021_

## Author Response (AR1)

We thank both referees for the valuable comments very and suggestion made to improve our manuscript. Below first, the suggestions of referee#1 and then to referee#2 (**bold**) and our responses to the individual points (normal font) in black as well as our changes made in the manuscript in blue (normal font) can be found:

**Ref#1: Abstract:**

1. **You should give a bit more information about the simulation set-up. For what period/condition did you run the spin-up?**

With the model, first, we conducted a 7000 year spin-up simulation with potential natural vegetation. This is followed by 390 years land use spin-up period with static cropland distribution pattern, all other crop and soil management as for the year 2010 (except for default tillage and bare soil management, Sect. 2.2; S 1.3, 1.4). With this spin-up we aimed to bring C and N pools in a dynamic equilibrium prior to the simulations with contrasting cover crop and tillage practices for the 50 year period.

We rephrased the abstract (also see responses and text edits described further below).

2. **Did you do a transient run without cover crops before the 50 year period with cover crops?**

Yes. During the first potential natural vegetation spin-up simulation period no cropland use dynamics are modeled at all. At the beginning of the subsequent land use (LU) spin-up period we simulated the one time LU conversion of natural ecosystem area to cropland distribution of the year 2010. We assumed that bare soil fallowing practices on cropland during main crop off-season periods as well as tillage on an annual basis on the entire cropland (for seedbed preparation). Cropland distribution as well as all further crop and soil management practices (fertilizer, manure) were held constant at the 2010 level during the entire LU spin-up period.

We rephrased the abstract (also see responses and text edits described further below).

3. **What does that 50 year simulation period represent, actually 50 years of the historical period to present day, 50 years of projection into the future, or do you simply loop over some years of climate forcing?**

The 50-year simulation period represents historical climate for 1962-2011 and all stylized management scenarios (with cover crops and/or no-tillage) are introduced from the year 1962 onwards. Therefore, the first decade after the introduction of the practice is not directly comparable to the last decade (in terms of weather) but we assume this error to be small, given that we average over larger areas when representing the results. We will add this point to the discussion.

We improved the description of the used model input data in Sect.2. and extended the discussion in Sect.4.5 on the used dynamic climate data to force our model simulations.

4. **Do your simulations account for changes in atmospheric CO2 which would affect the soil C uptake as well?**

Yes, our simulations include annual dynamic input data on $CO_2$ concentration levels. We agree that rising CO2 levels affect soil C uptake, which, in the same way as the transient climate, impedes the direct comparison between the first and the last decade. However, some of the field experiments to which we compare our results have been conducted over comparable time periods (up to 54 years) and are therefore affected by increasing CO2 levels, as well. We will add this point to the discussion.

We included the description of the used model input data in Sect.2. and extended the discussion Sect. 4.5 on the here used dynamic $CO_2$-concentration data which affect soil carbon dynamics in the model as well as can be assumed impacting field trials as well.

5. **These are all important details that you should shortly mention in the abstract.**

We will add a very brief summary of the simulation set up details to the abstract. However, since the description requires a lot of information, this can be only in very aggregated form. Beyond that, we will consider moving parts of Sect. S1.2 and S1.3 to the main text in Sect. 2.2., so that the reader can grasp applied methods and data in a quicker and more comprehensive way.

We added details to the simulation set up and rephrased the abstract. Further, we integrated paragraphs from the Supplement and extended the description of used methods, data, and simulation setup in Sect. 2 of the MS.

**Ref#1: Introduction:**

1. **The introduction is overall quite well written. I miss however a bit the connection between the last paragraph of the introduction, which lists the specific objectives of the study, and the rest of the introduction section.**

We will improve the connection between the discussion of the state-of-the- art in modeling carbon and nitrogen dynamics in croplands and our objectives, to better clarify our motivation.

We added several sentences in the introduction Sect. 1 highlighting our motivation, knowledge gaps in previous modeling assessments, and resulting connection to the objectives of our study.

2. **It would be good if you could work out some existing research gaps that you could fill with your study, and maybe formulate at least one major, overarching research question. (I guess that research question will ask for a global scale, quantitative assessment of the potential of cover crops to increase C storage and reduce N leaching, for a certain number of globally important crops, accounting for differences in climate and soil.)**

We will improve the last paragraph of Sect. 1 on objectives and add there: "The analysis is guided by the following research questions: 1. Which potential have cover crop practices to increase soil C storage and reduce nitrogen loss through leaching from global cropland to improve agroecosystem services and functions? 2. What effects on the productivity yields of following main crop can be expected from cover crop practices?"

We reformulated text in Sect. 1 including a paragraph on the aim of our study.

3. **In that context, it would also be good to summarize a bit the potential of land surface models to answer such a research question. In that context, you should also give the state of the art of the use of land surface models for this kind of research (in a broader sense: impact of agricultural land management on C and N cycles), and highlight what is new in your study compared to older studies applying land surface models, and in particular to studies (maybe also of your group) using older versions of LPJmL.**

We will extend the introduction by an overview of the state-of-the-art of applying global vegetation and land surface models to assess impacts of agricultural management practices on C, N, and water dynamics, e.g. citing Pongratz et al. (2018) and McDermid et al. (2017). For the comparison to studies using older versions of LPJmL we will refer to Lutz et al. (2019) and Herzfeld et al. (2021). We will also review the state-of-the-art of modelling the role of cover crops referring e.g. to Olin et al. (2015) and Kollas et al. (2015).

We added several paragraphs elaborating on the state-of-the-art of modeling assessments of cropland management impacts on states and processes within agroecosystems using land surface and dynamic vegetation models, referring to several other modelling studies' findings.

4. **In the last paragraph with the specific objectives, you should also mention that you only do simulations for herbaceous cover crops.**

We will add in the last paragraph of the introduction that in our analysis we focus on effects of non-leguminous cover crops, which are simulated as grasses.

We added the description and also reformulated the last paragraph of the introduction Sect. 1.

Own considerations:
We corrected the sentence on types of cover crop species referring to Florentín et al. (2011) in lines 65-66.

**Ref#1: Methods and data:**
1. **L93: "model skills" – you mean "model performance"?**

We will change the wording as suggested to 'model performance'.

We edited the sentence as suggested.

2. **L96-98: Why do you mention that? Did they do an evaluation of model performance that would support your study?**

We will modify the Herzfeld et al, 2021 sentence, saying that: they found good agreement with literature estimates on global simulated cropland soil carbon content in response to climate when accounting for historical dynamic land use change, cropland use and management practices. They use a similar model code (apart from

the here presented modified implementation of cover crops), and crop and management input data) so that we can refer to their model evaluation to describe the model performance of the model setup used here as well.

We modified and extended the text referring to Herzfeld et al. (2021) finding 170 PgC for global cropland soils in the year 2018 (and for the period years 200-2009) using LPJmL5.0-tillage2.

3. **L106-109: I am not sure why you are mentioning that. Is that to show a limitation in existing models that you might want to overcome? This is not clear to me. It also seems that this could be put into the introduction section, where you should outline the state of the art for this kind of model approaches (see my comment on the Introduction).**
Yes, indeed we want to say that many crop models, including the older versions of LPJmL used in Kollas et al. (2015) show limitations for this management aspects and we aim to overcome it. We will modify the text to highlight that this motivated our model development work in LPJmL for this study.

We moved that sentence referring to the study of Kollas et al. (2015) to the introduction Sect. 1. on state-of-the-art of modeling sustainable land management practices.

4. **L110-113: So you only use grass like cover crops? That should be mentioned already in the introduction. Do you use the three different grass types (tropical C4, temperate C3, and polar C3 grass) depending on the climate zone?**

Yes, we represent non-leguminous cover crops with LPJmL's three herbaceous PFTs (grasses). In the model, the establishment of the different grass types is determined by their bio-climatic limits, which characterize them as tropical C4, polar C3, or temperate C3 grass plants. We will revise the text to make this clearer (see also related comment above).

We modified the last paragraph of the introduction (Sect. 1) and text in Sect. 2.2 highlighting that we focus our analysis on effects of annual grass cover crop types.

5. **Section 2.2: Could you say something about your scenarios from fertilizer and manure application? Do you represent irrigation? Or is it all rain-fed agriculture?**

Following the request to better describe the simulation experiments, we will move and extend the sections on the modeling protocol and input data description from the supplementary to the main text. Irrigated and rainfed crop production systems are modeled per crop type and both can occur in the same grid cell. Their grid scale distribution is represented in the model simulations based on LUH2v2 (year 2010) by Hurtt et al. (2020). Also, mineral N fertilizer and manure are separate, gridded model input data.

We improved the description of the used model input data on nitrogen fertilizer, manure, as well as the representation of rainfed and irrigated cropping system dynamics in the model.

6. **L126-129: Did you do that spin-up for present day conditions, or for pre-industrial conditions (in particular with regard to atmospheric CO2)? That is not clear. Does atmospheric CO2 levels impact vegetation-soil carbon dynamics in your model? Which climate forcings did you use? What years do they represent? These are important details and should be mentioned here.**

As per the earlier request for more detail (above), we will also describe the spin-up simulations, as well as extend and improve the information on the climate input data (based on CRU for years 1901-2014, (Harris et al., 2014)) and atmospheric $CO_2$ concentrations (based on the times series from Mauna Loa station measurement data for years 1841-2011).

We moved the description of model input data from the Supplement to the main text Sect.2 and extended the description of climate and $CO_2$ data used in our simulation setup.

7. **L130-136: With that spin-up, did you bring the C and N stocks on cropland in steady state for conventional management practises? If so, I do not understand the sense of the first spin-up?**

Yes, we attempted with the two-step spin-up: to first bring the potential natural vegetation and soil C, N pools into equilibrium to mimic pre-anthropogenic conditions of the environment as a starting point. Then, we continued with a second spin-up in which land use is introduced with the 2010 pattern, so that the simulation years analyzed (after introducing different cover crop and tillage scenarios) start from cropland that has reached a new dynamic equilibrium, which we assume to be representative for field trials, which we compare to.

We added text in Sect. 2.4 highlighting that we aim to start from soils in equilibrium, before introducing alternative management practices in our simulations and to better match experimental field conditions reported in meta-analyses or reviews, which values we use for evaluating the implemented cover crop model code functionalities.

**Ref#1 Results:**

1. **Table 1: You only start to discuss that table in the discussion section. I would suggest to move this table there, as it mainly serves comparison between simulation results and literature values. Why don't you also show the results for CCNT is this table? In the last subsection of the results section, you present the statistics applying as mask the map of areas where conservation agriculture was actually applied during 1974-2010. I guess it would make much sense to use that mask also for this table here and the comparison between model results and literature values, in particular for the impact of cover crops on yields.**

We will follow the recommendation to move Table 1 to the discussion section. Table 1 merely summarizes CC results shown also in the graphs of the result section to compare them to literature values. Unfortunately, for a similar table for the evaluation of CCNT results we lack appropriate literature values. Although we briefly compare CCNT effects on crop yields to Pittelkow et al. (2015) in section 4.3, the effects of no-tillage in combination with crop rotation do not provide a good reference point for our effects of CCNT. Regarding the use of a conservation agriculture mask to calculate statistics in Table 1 see detailed comment below.

We moved the Table 1 and referring text to the discussion section. Further, we added respective tables for global aggregated responses of CCNT and NT per decade in the Supplement Sect. S2.4 and for estimates of the three alternative management practices on CA area in Sect. S2.6. In Sect. 4.5 we added text discussing the uncertainty on the presented estimates for historical management impacts on CA area, through 'partial adoption of CA' practices mostly referring to reduced tillage but not necessarily also the rotation and soil cover aspects.

2. **L256-259: I guess this is because in drier regions, you simulate irrigation, as you mention below. But in fact, in drier regions where water is limited, you might want to abstain from planting cover crops if they lead to additional evapotranspiration losses. You should come back to that thought in the discussion section.**

We will add a sentence in the discussion about the possible trade-off between yield gains and increased irrigation water demand due to cover crops in irrigated systems and dry regions.

We added details to the text in Sect. 4.4 and a paragraph for main crop yield effects in dry areas due to cover crops and to the discussion Sect. 4.5 highlighting the difference in irrigation water requirements assumed with cover crops in irrigated cropping systems.

3. **L274-278: Is that because under rain-fed conditions, cover crops increase the water limitation for the cash crops? What is the irrigation scenario: do you always fully irrigate or is there deficit-irrigation possible? Do you take into account the limited availability of irrigation water? Under irrigated conditions, are cover crops irrigated as well?**

In our simulation setup, irrigation is implemented to fully irrigate crops on irrigated cropland (no deficit irrigation). Cover crops are not irrigated at all. In the simulation for this study we did not account for limited water availability to assure unambiguous irrigation effects (removal of water stress). Therefore, the reason for this pattern must be related to a change in N availability from cover crops. We will come back to this in the discussion and improve the method section on irrigation.

We modified the description of cover crop model code functionalities in Sect. 2.2 highlighting that cover crops are grown always under rainfed conditions without irrigation - independent of the main crop irrigation setting. Further, we extended and moved the description of the rainfed, and irrigation pattern represented in the model simulations from the Supplement to the main text Sect. 2.1 and 2.2.

**Ref#1 Discussion:**

1. **L300-301: See my comment for table 1. It would make much more sense if you masked your results by the map of conservation agriculture. If you use the statistics including areas where conservation agriculture incl. cover crops is not yet applied (maybe because of water shortage that would be increased by cover crops), your simulation results are not comparable to empirical findings. Further, it would make more sense to also compare model results vs. observations per regions, and further distinguishing more clearly**

**rainfed and irrigated agriculture in that comparison. That seems very important as you have highlighted the huge spatial variations and a general difference between irrigated and rainfed agriculture before.**

The literature values to which we compare our CC results in Table 1 are reviews and meta-analyses, which quantify the effects of cover crops in tillage systems (conventional agriculture) and mainly for rainfed systems. Therefore, it would be inappropriate to calculate statistics for CA areas to compare to these literature values. We will improve the description of literature values in the manuscript to make this clear. Regarding the differentiation of regional cover crop impacts as well as in irrigated vs. rainfed systems, we will search for additional studies to include in the comparison.

See comments and added values above for ref#1-results-comment 1 on Table 1 that we add the idealized management scenario results for NT and CCNT per decade to the Supplement Table S2.3. and for the CA area to the Table S2.5.
We improved the description of values derived from the literature and used for the evaluation of the LPJmL model code functionalities for the representation of cover crop practices in Sect. 2.4 and 2.5 of the main text.
And we added to the discussion Sect. 4.3 that the majority of studies used for comparison in our study relate to rainfed systems effects of cover crops or do not provide detailed management information . Further, we included values referring explicitly to irrigated or rainfed systems from Lutz et al, 2019 for yield changes with simulated no-tillage, and Quemada et al. (2013) for N leaching and yield effects in irrigated cropping systems. We largely extended the discussion on irrigation and rainfed impacts in Sect 4.5.

2. **L307-310: For that upscaling exercise, why would you use the median rate and not the mean rate? The latter would seem more appropriate. At least you should try to justify using the median.**

We have chosen the median rate to better relate to the other results in the paper where the median showed larger robustness towards outlier values when aggregating gridded values to the global scale. However, we will add the means in the revised manuscript.

We added the mean rate and modified the text accordingly.

3. **Subsection 4.2: I wonder if you could disentangle the effects of increased N-uptake by vegetation vs. changes in runoff and drainage. You described before how cover crops increase the evapotranspiration, which should lead to a decrease in drainage and thus advective export of reactive N species. You mention that cover crops don't have a big effect on N leaching in dry regions. Might this be due to the fact that drainage is low in dry areas, and irrigation water is only applied to satisfy evapotranspiration requirements, with no excess water feeding additional drainage?**

Due to the complex mutual interactions between water fluxes, N fluxes, and plant growth in LPJmL, it is challenging to disentangle the contribution of N uptake by cover crops and drainage reduction to the reduction in N leaching. We will investigate this aspect and extend the discussion accordingly. At the very least we will improve the description of the different processes that lead to an increase in N leaching. Regarding the question on drainage from irrigation water: Depending on the irrigation system, more water than required by the plant is actually applied to the field, which results in enhanced drainage. The effect is pronounced for surface irrigation and weak for drip irrigation. We will mention this aspect in the discussion.

We improved general description of rainfed and irrigated cropping systems represented in the model simulations in the entire MS. Further, we added text to the discussion Sect. 4.3 on N leaching and water dynamics impacted by cover crops in dry regions with respect to drainage and irrigation.

4. **L384-386: Soybeans are also often irrigated, and would thus have no penalty from additional water consumption through cover crops. Could you investigate if that is the reason in your findings?**

We will investigate if this effect can be attributed to irrigation and extend the discussion accordingly.

We modified the discussion Sect. 4.4 on cover crop impacts highlighting the difference of effects in irrigated and rainfed cropping system, especially for irrigated soybean for which N and water competition with the cover crops are excluded in our simulations, due to the potential irrigation setting (unrestricted water availability for irrigation) and because soybeans as N-fixers are not affected by the lower N availability induced by cover crop cultivation in our simulations.
Further, we added information on the crop-specific area share of cropland to the caption of Fig. 5 and in S2.5 to indicate the relative contribution of irrigated system dynamics in averaged yield effects obtained with the alternative land management simulations compared to the reference simulations.

Own considerations:
We deleted a sentence in the discussion because it was mistakenly occurring twice.

*References*

Florentín, M. A., Peñalva, M., Calegari, A., and Derpsch, R.: Greeen manure/cover crops and crop rotation in Conservation Agriculture on small farms, Food and Agriculture Organisation of the United Nation (FAO), Rome, 2011.

Harris, I., Jones, P. D., Osborn, T. J., and Lister, D. H.: Updated high-resolution grids of monthly climatic observations – the CRU TS3.10 Dataset, International Journal of Climatology, 34, 623-642, doi: https://doi.org/10.1002/joc.3711, 2014.

Herzfeld, T., Heinke, J., Rolinski, S., and Müller, C.: Soil organic carbon dynamics from agricultural management practices under climate change, Earth System Dynamics, 12, 1037-1055, doi: 10.5194/esd-12-1037-2021, 2021.

Hurtt, G. C., Chini, L., Sahajpal, R., Frolking, S., Bodirsky, B. L., Calvin, K., Doelman, J. C., Fisk, J., Fujimori, S., Klein Goldewijk, K., Hasegawa, T., Havlik, P., Heinimann, A., Humpenöder, F., Jungclaus, J., Kaplan, J. O., Kennedy, J., Krisztin, T., Lawrence, D., Lawrence, P., Ma, L., Mertz, O., Pongratz, J., Popp, A., Poulter, B., Riahi, K., Shevliakova, E., Stehfest, E., Thornton, P., Tubiello, F. N., van Vuuren, D. P., and Zhang, X.: Harmonization of global land use change and management for the period 850–2100 (LUH2) for CMIP6, Geoscientific Model Development, 13, 5425-5464, doi: https://doi.org/10.5194/gmd-13-5425-2020, 2020.

Kollas, C., Kersebaum, K. C., Nendel, C., Manevski, K., Müller, C., Palosuo, T., Armas-Herrera, C. M., Beaudoin, N., Bindi, M., Charfeddine, M., Conradt, T., Constantin, J., Eitzinger, J., Ewert, F., Ferrise, R., Gaiser, T., Cortazar-Atauri, I. G. d., Giglio, L., Hlavinka, P., Hoffmann, H., Hoffmann, M. P., Launay, M., Manderscheid, R., Mary, B., Mirschel, W., Moriondo, M., Olesen, J. E., Öztürk, I., Pacholski, A., Ripoche-Wachter, D., Roggero, P. P., Roncossek, S., Rötter, R. P., Ruget, F., Sharif, B., Trnka, M., Ventrella, D., Waha, K., Wegehenkel, M., Weigel, H.-J., and Wu, L.: Crop rotation modelling—A European model intercomparison, Eur. J. Agron., 70, 98-111, doi: http://dx.doi.org/10.1016/j.eja.2015.06.007, 2015.

Lutz, F., Herzfeld, T., Heinke, J., Rolinski, S., Schaphoff, S., Von Bloh, W., Stoorvogel, J., and Müller, C.: Simulating the effect of tillage practices with the global ecosystem model LPJmL (version 5.0-tillage), Geoscientific Model Development, 12, 2419-2440, doi: https://doi.org/10.5194/gmd-12-2419-2019, 2019.

McDermid, S. S., Mearns, L. O., and Ruane, A. C.: Representing agriculture in Earth System Models: Approaches and priorities for development, Journal of Advances in Modeling Earth Systems, 9, 2230-2265, doi: 10.1002/2016MS000749, 2017.

Olin, S., Lindeskog, M., Pugh, T. A. M., Schurgers, G., Wårlind, D., Mishurov, M., Zaehle, S., Stocker, B. D., Smith, B., and Arneth, A.: Soil carbon management in large-scale Earth system modelling: implications for crop yields and nitrogen leaching, Earth System Dynamics, 6, 745-768, doi: https://doi.org/10.5194/esd-6-745-2015, 2015.

Pittelkow, C. M., Liang, X., Linquist, B. A., van Groenigen, K. J., Lee, J., Lundy, M. E., van Gestel, N., Six, J., Venterea, R. T., and van Kessel, C.: Productivity limits and potentials of the principles of conservation agriculture, Nature, 517, 365-368, doi: https://doi.org/10.1038/nature13809, 2015.

Pongratz, J., Dolman, H., Don, A., Erb, K.-H., Fuchs, R., Herold, M., Jones, C., Kuemmerle, T., Luyssaert, S., Meyfroidt, P., and Naudts, K.: Models meet data: Challenges and opportunities in implementing land management in Earth system models, Global Change Biology, 24, 1470-1487, doi: 10.1111/gcb.13988, 2018.

Quemada, M., Baranski, M., Nobel-de Lange, M. N. J., Vallejo, A., and Cooper, J. M.: Meta-analysis of strategies to control nitrate leaching in irrigated agricultural systems and their effects on crop yield, Agriculture, Ecosystems and Environment, 174, 1-10, doi: https://doi.org/10.1016/j.agee.2013.04.018, 2013.

Below the suggestions of referee#2 (**bold**) and our responses to the individual points (normal font) in black and in blue our referring changes made in the MS (normal font) can be found:

**Ref#2:**

1. **My main comment is with the presentation/organization of the paper. There are three main model simulations: 1) cover crops, 2) no till, and 3) both. Yet, these three simulations could be presented in a more consistent way. For example, Table 1 describes the soil C sequestration of cover crops only. I was also expecting to see analogous tables for tillage and CCNT.**

In the main text, we focus on the analysis of cover crops effects (CC) and their sensitivity to tillage and no-tillage (CCNT). The results for the no-till scenario (NT) are merely indented to support the interpretation of differences between the CC and CCNT scenarios. To improve the balance between CC and CCNT results in the paper, we will shift Figure S2.5 to the main text (as also requested below) and improve the presentation and discussion of CCNT results where needed. To address the request by referee#1 we will move Table 1 to the discussion as it solely provides a summary of CC results for comparison and evaluation of model results to literature estimates.
Since we lack corresponding literature values for CCNT we cannot provide an analogous table for CCNT results. However, for comprehensiveness we will add overview tables similar to Table 1 but without literature values for CCNT and NT to the supplement.

We moved Table 1 to the discussion Sect. 4.1 on evaluation of modified code functionalities and simulation of cover crop practices using LPJmL5.0-tillage-cc.  Further, we added similar tables as former Table 1, with global aggregation of CCNT and for NT per decade to the Supplement 2.5 and are referring to it in Sect 4.1. Then, we moved the former Fig.S2.5 from Supplement to the main text, now as Fig. 6 for yield response pattern to CCNT for the four analyzed main crop types.

2. **If similar literature estimates have already been provided in previous work on LPJml-tillage2, then perhaps the authors could refer to that, but CCNT is a new interaction that has not yet been modeled with LPJml, so I think that it merits a comparison to observed values.**

Unfortunately, we were not able to find appropriate literature values to validate our CCNT results directly. Therefore, we have also provided the results for the no-tillage setting (NT scenario), which has been extensively validated in Lutz et al. (2019) for a previous LPJmL code version. Together with the here presented evaluation of the CC results obtained with the model, we assume this to be the best available option to support the interpretation of CCNT effects. We will clarify the role of the NT scenario in the revised manuscript and improve its utilization for the interpretation of CCNT results, especially regarding soil N and water effects.

We extended the MS, i.e. Sect 4.2 and 4.4 discussing our simulated no-tillage results in relation to finding of Lutz et al. (2019), who assessed no-tillage effects compared to conventional tillage practices using LPJmL5.0-tillage.

3. **Again, Figure 4 only shows productivity response to cover crops, while all of the other figures show all three model simulations.**

As requested below we will move S2.5 to main text and will add the 4 maps for productivity impact of NT into the SI.

We moved the referring plot from previous Fig. S2.5 to main text, there now as Fig. 6.

4. **I would also be interested to know if LPJml predicts a similar total C stock (and maybe yield, veg C, GPP, NPP..) as LPJml-tillage2 from Herzfeld et al. (2021) or LPJml4 (Schaphoff et al. 2018b), basically to show if the model is indeed similar except for these new features.**

LPJmL5 fundamentally differs from LPJmL4 in that it includes a mechanistic representation of the N-cycle. Resulting differences in model behavior are presented and discussed in von Bloh et al. (2018). Herzfeld et al. (2021) uses a very similar model code to ours, which differs mainly in the implementation of cover crops. However, the results presented in Herzfeld et al. (2021) are not entirely comparable to ours, because we use another simulation protocol. We will include the reference to von Bloh et al. (2018) in the revised manuscript and point out the similarity between model versions used in Herzfeld et al. (2021).

In the Introduction we refer to findings of Herzfeld et al. (2021) which motivated our study and in Sect. 2.1 we extended the description of their used model code version compared to ours and stating their SOC value for cropland. In Sect. 4.2 and Sect. 4.4 we added results found in Lutz et al. (2019) on soil carbon and yield with no-tillage using a similar LPJmL model versions as well as added our results for the comparison.

**5. L54: A word is missing, maybe: "by this [method] may"**

Yes, we will modify the text to '...and in this way may reduce fertilizer...' .

We rephrased the sentence in the MS.

**6. L67: Change "glass" to "grass"**

We will rephrase the sentence, because here we refer to area covered with green houses made of 'glass' in contrast to bare soil or vegetated ground cover- it is the wording for categories used in the cited EUROSTAT statistics on 'soil cover'.

We added the term 'greenhouses' to the sentence in the MS.

**7. Mention somewhere in the methods that LPJml simulates all of the crops mentioned in TablS1.1 but that in this paper, you only focus on maize, rice, wheat and soybean. I think it is fine to focus on the four major crops, but it is worth highlighting to readers that there are others. It is also worth noting if they are included in any crop averages or totals reported in the paper.**

We will improve Sect. 2, adding that we do model all crop types as indicated in the Table S1.1. Further, we add that we focus the productivity impact analysis on the four crop types because of their global relevance as staple crops for food security and their large shares on cropland. We will also clarify that results for soil C and N leaching always refer to the entire cropland within a grid cell.

We added the aspect to the MS, Sect. 2.5 and extended the description of cropland data used for aggregation regarding the area-weighting of either crop-specific results with crop-specific area for the productivity analysis, or area-weighting with the sum of physical cropland per grid cell for soil C sequestration and changes in soil N leaching rates.

**8. Figure 5: It would be useful to know the number of grid cells (or whatever spatial unit is being used) in Figs 5, S2.4.1, and S2.4.2. From looking at the three graphs together, it looks as though most locations are rainfed rather than irrigated, and that the small response and variability of the irrigated locations could also be due to the lower sample size.**

We will add the corresponding numbers of grid cells and area covered by each crop type (and per water regime) to the graphs in Fig 5 and Fig. S2.4.

We added the sums of crop-specific area in hectares, as well as of single irrigated and rainfed shares to the figures in the MS, including for Fig. 5, and Fig. S2.5.

**9. L315: Here the authors mention that there is a time lag in response of soil C sequestration rates, and while perhaps one could detect a change in soil C using a model, I would not expect field measurements to reflect soil C changes for at least a few years, due to the relatively small signal in such a large pool.**

We agree. Also the authors cited in that paragraph, e.g. (Blanco-Canqui et al., 2015; Laborde et al., 2020; West and Six, 2007) in their reviews and meta-analysis do find delayed responses on soil C and yields after switching to cover crop or generally to another soil management practice. We will mention the challenge to detect changes in soil C and other variables in the first years after switching to another soil management practice in field experiments.

We reformulated the referring sentence in the MS.

**10. L335: Could be useful to know how you define equilibrium.**

Our assumption on 'equilibrium' as effect detected in our alternative management simulations are based on Poeplau and Don (2015), who defined "the new steady state to be reached after the annual change in SOC stock fell below 0.01 Mg ha-1 yr-1." We will add this definition in the revised manuscript.

We added the definition of equilibrium referring to Poeplau and Don (2015) to the Sect.4.2.

**11. L463: I would say "model prediction" instead of "quantification" here. In general, it would be good to use language recognizing that these are model predictions and not measurements.**

Thanks for hinting to this, we will improve the wording in the entire MS to make clearer and harmonize for model 'projections' or 'simulation' results.

We rephrased the referring paragraph and modified several sentences in the MS specifying that we refer to simulated management scenarios results.

**12. L484: Perhaps "conclude" instead of "resume"?**

We will improve for 'conclude' instead of 'resume'.

Done.

**13. Table S2.1: It is interesting that yields tend to increase with CC for specific crops, but in non-legume averages, yield losses tend to be larger than the modeled losses. Why do you think the meta-analyses disagree with the national statistics?**

We will add to the discussion for model evaluation that values derived at field scale measurements under highly controlled conditions may reflect local conditions rather than covering the variance of environmental and socio-economic conditions captured with the global model set-up applied here.

We added a paragraph in Sect. 4.5 elaborating on differences of local specific field trial differences versus modeled gridded and global scale effects reported in our study.

**14. Figure S2.5: I know there are already a lot of figures in the main manuscript, but this one seems as important as cover crops to the paper's main conclusions.**

We will move the Fig. S2.5 to the main text.

Done.

**15. Optional, just a thought: It would be interesting to see if the C that is "lost" as a result of a reduction in yield is proportional to the C gained in the soil. It seems for example, that land management practices with less yield loss (like NT) also have less soil C gain.**

Cover crops have a stronger impact on water, carbon, and nutrient cycles than no-tillage alone, which leads to smaller effects, in general. Yield changes in the CC and CCNT scenarios are very heterogeneous across crop and do not appear linked directly to changes in soil C. Nevertheless, we will investigate these aspects in our analysis if appropriate. Soil carbon dynamics are additionally determined by the spatial pattern of the crop type, the crop specific growing season length, fertilizer (Fig S1.2b), and other crop management modeled at the grid scale, which we will emphasize more in the discussion.

We added a paragraph elaborating on this aspect in more detail in the discussion Sect. 4.5. Our study presents results for a variety of agroecosystem variables to cover crop impacts, discussing their causes, as well as the process based biophysical and biogeochemical processes. We still are convinced that the suggested analysis on the trade-offs are an interesting research question. However, the model simulations used here do not allow for separated effects of soil processes per crop type because in our model after the end of the main crop growing season soil columns are merged, preserving their relative cropland shares to averages only separated in rainfed and irrigated fractions. This way effects are not attributable entirely per crop type for dynamics in soil carbon or changes in nitrogen leaching rates but would require single crop management scenario simulations but also the assessment of associated emissions, which we see beyond the scope of our analysis. We specified the

information on the set-aside dynamics in Sect. 2.2. Further we elaborate on this challenge in an added paragraph in Sect. 4.5.

**References**

Blanco-Canqui, H., Shaver, T. M., Lindquist, J. L., Shapiro, C. A., Elmore, R. W., Francis, C. A., and Hergert, G. W.: Cover Crops and Ecosystem Services: Insights from Studies in Temperate Soils, Agronomy Journal, 107, 2449-2474, doi: https://doi.org/10.2134/agronj15.0086, 2015.

Herzfeld, T., Heinke, J., Rolinski, S., and Müller, C.: Soil organic carbon dynamics from agricultural management practices under climate change, Earth System Dynamics, 12, 1037-1055, doi: 10.5194/esd-12-1037-2021, 2021.

Laborde, J. P., Wortmann, C. S., Blanco-Canqui, H., Baigorria, G. A., and Lindquist, J. L.: Identifying the drivers and predicting the outcome of conservation agriculture globally, Agricultural Systems, 177, 102692, doi: https://doi.org/10.1016/j.agsy.2019.102692, 2020.

Lutz, F., Herzfeld, T., Heinke, J., Rolinski, S., Schaphoff, S., Von Bloh, W., Stoorvogel, J., and Müller, C.: Simulating the effect of tillage practices with the global ecosystem model LPJmL (version 5.0-tillage), Geoscientific Model Development, 12, 2419-2440, doi: https://doi.org/10.5194/gmd-12-2419-2019, 2019.

Poeplau, C. and Don, A.: Carbon sequestration in agricultural soils via cultivation of cover crops – A meta-analysis, Agriculture, Ecosystems and Environment, 200, 33-41, doi: https://doi.org/10.1016/j.agee.2014.10.024, 2015.

von Bloh, W., Schaphoff, S., Müller, C., Rolinski, S., Waha, K., and Zaehle, S.: Implementing the nitrogen cycle into the dynamic global vegetation, hydrology, and crop growth model LPJmL (version 5.0), Geoscientific Model Development 11, 2789-2812, doi: https://doi.org/10.5194/gmd-11-2789-2018, 2018.

West, T. O. and Six, J.: Considering the influence of sequestration duration and carbon saturation on estimates of soil carbon capacity, Climatic Change, 80, 25-41, doi: https://doi.org/10.1007/s10584-006-9173-8, 2007.